# Innovative Seafood Preservation Technologies: Recent Developments

**DOI:** 10.3390/ani11010092

**Published:** 2021-01-06

**Authors:** Michael G. Kontominas, Anastasia V. Badeka, Ioanna S. Kosma, Cosmas I. Nathanailides

**Affiliations:** 1Department of Chemistry, University of Ioannina, 45110 Ioannina, Greece; abadeka@uoi.gr (A.V.B.); i.kosma@uoi.gr (I.S.K.); 2Department of Agriculture, University of Ioannina, 47100 Arta, Greece; nathan@uoi.gr

**Keywords:** seafood spoilage, food safety, high hydrostatic pressure, natural preservatives, ozonation, irradiation, pulse light technology, retort pouch processing

## Abstract

**Simple Summary:**

The present review summarizes the findings of recent studies on innovative seafood processing technologies including high hydrostatic pressure, natural preservatives, ozonation, irradiation, pulse light technology and retort pouch processing as well as referencing the legal aspects pertaining to these technologies.

**Abstract:**

Fish and fishery products are among the food commodities of high commercial value, high-quality protein content, vitamins, minerals and unsaturated fatty acids, which are beneficial to health. However, seafood products are highly perishable and thus require proper processing to maintain their quality and safety. On the other hand, consumers, nowadays, demand fresh or fresh-like, minimally processed fishery products that do not alter their natural quality attributes. The present article reviews the results of studies published over the last 15 years in the literature on: (i) the main spoilage mechanisms of seafood including contamination with pathogens and (ii) innovative processing technologies applied for the preservation and shelf life extension of seafood products. These primarily include: high hydrostatic pressure, natural preservatives, ozonation, irradiation, pulse light technology and retort pouch processing.

## 1. Seafood: Definitions, Structure and Composition

The term “seafood” includes (i) free-swimming, pelagic and freshwater fish, (ii) crustaceans, (iii) mollusks and (iv) the respective aquacultured species. The most important crustaceans include: crabs, lobsters, crayfish, shrimp and prawns, while the most important mollusks include: mussels, scallops, cockles, oysters, clams, squid and octopus. Fish may be further categorized as saltwater vs. freshwater fish, fatty vs. non-fatty fish and free-swimming vs. aquacultured fish. Although the structure and composition of fish are similar to those of meat, the former bear distinctive features. Firstly, fish have no obvious deposit of fat. Even though the fat content of fish may exceed 25% *w*/*w*, it is mostly dispersed within muscle fibers. A second feature is that the connective tissue is usually less than 3% as opposed to meat in which the connective tissue content may be as high as 15%. A third feature is fish muscle structure. Animal muscle is made up of long fibers as opposed to fish in which muscle fibers form short segments known as “myotomes”. Such fibers are separated by sheets of connective tissue known as “myocommata” [1]. This is mainly responsible for the flaky texture of fish flesh. A fourth specific feature of fish muscle is the presence of non-protein nitrogenous compounds composed of free amino acids, volatile nitrogen bases such as ammonia, trimethylamine and trimethylamine oxide (TMAO), creatine, taurine, uric acid, anserine, carnosine and histamine. According to Shewan [2], non-protein soluble components constitute ca. 1.5% of fish muscle.

The composition of fish muscle varies significantly with species, size, fishing grounds and diet, especially in the case of aquacultured fish [3]. The typical composition for a non-fatty fish such as cod would be: moisture 81.5%, protein 16.5%, fat 0.4%, carbohydrate 0%, ash 1.2%. The respective composition for a fatty fish such as salmon would be: moisture 63.5%, protein 17.5%, fat 16.5%, carbohydrate 0%, ash 1%. Fish flesh can also contain non-protein nitrogenous compounds, typical examples being those of cod (total N = 2.83%, protein N = 2.47%) and lobster (total N = 2.72%, protein N = 2.04%). The proximate composition of selected seafood products is presented in Table 1.

## 2. Seafood Spoilage

As with the flesh of terrestrial animals and birds, the muscular tissue of fresh caught fish is normally sterile at harvest. On the contrary, skin, gills and intestines carry a large number of bacteria (10^2^–10^7^ cfu/cm^2^ on skin and 10^3^–10^9^ cfu/g in gills and gut). The spoilage microflora of fresh ice-stored fish consist mainly of Gram-negative *Pseudomonas* spp. and H_2_S-producing bacteria including *Shewanella putrefaciens*. *Acinobacter* and *Moraxella* spp. may comprise a portion of the spoilage microflora [1]. Olafsdottir et al. [5] reported on the spoilage microflora of haddock fillets stored at refrigeration and abuse temperatures and reported *Photobacterium phosphoreum* to be the dominant spoilage microorganism. *Pseudomonas* spp. and *Shewanella putrefaciens* were also present, being responsible for spoilage off-odors. The early stages of spoilage involve utilization of non-protein nitrogen, resulting in the formation and accumulation of fatty acids, ammonia and volatile amines. As proteolysis proceeds, spoilage becomes more evident. Hydrogen sulfide and other sulfur compounds, such as mercaptans and dimethyl sulfide, produced by *S. putrefacians* and some pseudomonads usually contribute to spoilage [3].

After harvesting from the fish farm or capture at sea, fish may either be stored in ice or fresh-frozen. The flesh of mollusks differs from that of crustaceans and free-swimming fish in that it contains an appreciable amount of carbohydrates in the form of glycogen. Even though microorganisms involved in mollusks’ spoilage are the same as those encountered in fish and crustaceans (*Pseudomonas* and *Acinetobacter-Moraxella* spp.), spoilage of the former is primarily glycolytic (it contains 1–5% glycogen) rather than proteolytic, leading to a pH decrease from around 6.5 to 5.8. Under such conditions of acidity, enterococci, lactobacilli and yeasts dominate the later stages of spoilage.

On the other hand, crustaceans such as shrimp and prawns, in addition to their endogenous microflora, are subject to a more rapid microbiological spoilage due to usual contamination with bacteria from the mud trawled up along with these species following capture [1]. Microbial spoilage of crustaceans occurs in a similar manner to fish flesh; however, the higher amount of free amino acids and other soluble nitrogenous compounds present leads to rapid spoilage and elevated levels of volatile basic nitrogen spoilage compounds [3]. More specifically, fish spoilage can be attributed to (i) post-mortem enzymic autolysis, (ii) microbial growth and (iii) oxidation of lipids.

### 2.1. Spoilage Due to Autolytic Enzyme Activity

Immediately after slaughter, post-mortem biochemical changes occur as a result of autolytic enzyme activity in fish flesh [6]. Autolytic enzymes present in seafood products tend to be highly active; seafood begins to undergo autolysis more rapidly than terrestrial animals. This is due to the fact that pH in seafood typically does not decline to the level of terrestrial muscle. Autolysis mainly affects the texture of fish muscular tissue during the early stages of deterioration but does not produce any characteristic spoilage off-odors and off-flavors [7]. As a result, the shelf life of fish can be restricted by the activity of these enzymes, even when spoilage microorganisms are not yet dominant. Autolytic enzymes of muscle and visceral organs can drive proteolysis during processing and storage of whole fish, leading to protein decomposition and solubilization. Such degradation also produces peptides and free amino acids via autolysis of fish muscle proteins, as well as biogenic amines through the action of decarboxylases, leading to the spoilage of fish meat [8]. According to Liston [9], proteases and lipases play a key role in protein and lipid degradation, respectively, during fish spoilage.

The degree of freshness of fish can be evaluated using numerous biochemical indices, i.e., the K-value, an index based on the rate of adenine nucleotides’ catabolism. Post-mortem changes in the K-value correlate well with the level of spoilage. The K-value starts off with a low value around 10% which increases along with endogenous enzyme activity. It, then, exponentially increases as microbial activity takes over the spoilage mechanism [10]. A K-value < 20% corresponds to optimal fish freshness. A 20 < K < 40% corresponds to acceptable fish freshness, while a K > 40% corresponds to unacceptable fish quality. Ahmad et al. [11] used microbiological, chemical and physical indices to evaluate the quality of sea bass fillets packaged in gelatin films containing 25% lemongrass essential oil and stored at 4 °C. The controls were considered unacceptable (K-value: 66%) on day 10, while treated samples did not reach this value on day 12 of storage.

### 2.2. Microbial Spoilage 

Depending on the specific aqueous environment, the microbial load of fish includes *Pseudomonas*, *Alcaligenes*, *Vibrio*, *Serratia* and *Micrococcus* spp. [12]. Microbial growth and enzyme activity are the major causes of fish spoilage, producing a large number of unpleasant off-flavors due to the formation of specific alcohols (mainly ethanol), aldehydes, acids, ketones and sulfur and nitrogen compounds [13]. Unpreserved fish are usually spoiled as the result of growth of Gram-negative, fermentative bacteria (such as Vibrionaceae), whereas chilled fish are spoiled by psychrotrophic Gram-negative bacteria such as *Pseudomonas* spp. and *Shewanella* spp. It is, therefore, important to distinguish between non-spoilage microflora enumerated within the total volatile count (TVC) and specific spoilage bacteria (SSO) as it is the latter that are responsible for fish spoilage [14]. The proliferation of bacteria leads to protein breakdown and the production of trimethylamine (TMA). For this reason, the levels of TMA are another widely used indicator to determine the extent of fish spoilage. Ammonia-like off-flavors are produced by TMA which is formed via the reduction of trimethylamine oxide (TMAO), a compound used by a number of spoilage bacteria such as *Shewanella putrefaciens*, *Aeromonas* spp., psychrotolerant Enterobacteriacceae, *P. phosphoreum* and *Vibrio* spp. to obtain energy [15]. At this point, it should be noted that the TMAO content of different families of marine species varies significantly. Another parameter of seafood freshness is total volatile basic nitrogen (TVB-N). TVB-N includes ammonia dimethlamine and trimethylamine, being formed through the decarboxylation of amino acids produced via protein degradation. Values between 30 and 45 mg TVB-N/100 g have been proposed as the upper limit of acceptability for various seafood products by Connell [15]. Besides the traditional chemical methods (TVB-N and TMA) to monitor microbial activity in fish, volatile organic compounds (VOCs) have also been more recently used as potential spoilage/freshness indicators [16]. Among the methods used to determine VOCs, solid-phase microextraction coupled to gas chromatography/mass spectrometry (SPME-GC/MS) has been successfully used to evaluate the degree of seafood spoilage [17]. *Pseudomonas fluorescens, Shewanella putrefaciens* and other spoilage bacteria grow rapidly during the intermediate stages of spoilage, which through their metabolism produce proteases and lipases responsible for the breakdown of proteins and lipids, respectively. In the later stages of spoilage, further growth of spoilage bacteria leads to the organoleptic rejection of fish [18]. It is important to note that fish muscle decomposition does not necessarily coincide with the presence of pathogens or toxins.

### 2.3. Oxidation and Hydrolysis

Oxidation of lipids is particularly important in fatty fish species such as mackerel, salmon and sardines. Lipid oxidation involves the reaction of unsaturated fatty acids of fish triglycerides with atmospheric oxygen to form hydroperoxides (primary oxidation products) which are unstable and therefore decompose to carbonyl compounds such as aldehydes and ketones (secondary oxidation products), responsible for characteristic rancid off-flavors. Molecular oxygen needs to be activated to singlet oxygen in order to allow oxidation to occur. Transition metals are primary activators of molecular oxygen [19].

Lipid hydrolysis in fish proceeds via enzymic or non-enzymic routes. The enzymic hydrolysis of fats by lipases is known as lipolysis. In this reaction, lipases break down triglycerides, producing free fatty acids, which leads to an increase in fish oil acidity and consequently a reduction in oil quality. Lipolytic enzymes can either be endogenous of the fish itself or can be the product of the psychrotrophic microorganism metabolism [20]. Lipases present in the fish skin, blood and tissue include triacyl lipase, phospholipase A_2_ and phospholipase B. The fatty acids formed during hydrolysis of fish lipids interact with sarcoplasmic and myofibrillar proteins, causing denaturation.

As previously mentioned, lipid oxidation involves the reaction of atmospheric oxygen with the double bonds of unsaturated fatty acids on the fish triglyceride molecules for the production of hydroperoxides. This initial reaction is catalyzed by hematin compounds (hemoglobin, myoglobin and cytochrome) [9,21].

Besides spoilage microorganisms, seafood is associated with many food pathogens. *Salmonella* spp. is the main cause of seafood-borne bacterial illness [1]. *Salmonella* spp. grow at temperatures between 5.2 and 47 °C and pH values between 3.7 and 9.5. However, numerous *Salmonella* strains can survive under freezing conditions for up to 9 months [22]. Even though *Listeria monocytogenes* is less widespread than *Salmonella* spp., it commonly occurs in seafood, being an environmental inhabitant of processing facilities. Like *Salmonella* spp., *L. monocytogenes* also grows between 0.4 and 45 °C and reaches high populations, particularly in shrimp and catfish. Among *Vibrio* spp., usually associated with gastroenteritis, the most virulent species is *Vibrio vulnificus*. Schwarz [23] applied rapid cooling to oysters and showed a 97.8% reduction in the population of *V. vulnificus*, whereas commercially cooled oysters took four days to reach the same value. *Clostridium botulinum*, a spore-forming anaerobe growing at temperatures exceeding 3.3 °C, is of main concern to the seafood industry because of its heat-stable, highly toxic toxin, produced during seafood storage. *Aeromonas hydrophila* is more commonly associated with seafood, particularly finfish and prawns. Papadopoulou et al. [24] showed that local seafood, 24 h after harvesting, contained *A. hydrophila* as the predominant organism (38% in freshwater fish; 73–86% in shellfish; and 93% in marine finfish).

## 3. Innovative Seafood Preservation Methods

The flesh of fish is composed of macroconstituents: moisture, proteins and fats, and microconstituents: minerals, vitamins and enzymes. In addition, crustaceans and mollusks contain carbohydrates in the form of glycogen. Due to their specific composition, seafood products are considered a very perishable commodity. The fact that fishing vessels gear seafood usually at large distances from the sites of consumption necessitates proper preservation to avoid product spoilage. This need is further driven by consumer demand for high-quality, lightly processed products with minimal changes in nutritional and sensory properties. This also applies to aquacultured seafood species which need to be properly preserved in order to be safely shipped to far away destinations. Besides traditional seafood preservation methods including chilling (at 0–1 °C), freezing (<1 °C), drying, smoking, salting, fermentation and canning, more recent methods of seafood preservation include (1) the use of natural preservatives, (2) high hydrostatic pressure treatment, (3) ozonation, (4) irradiation, (5) pulse light technology, (6) retort pouch processing and (7) packaging in combination with refrigeration or freezing [25,26,27,28,29,30,31].

### 3.1. Use of Natural Preservatives

Fresh seafood products are extremely perishable even when refrigerated and susceptible to microbial growth, autolytic activity and lipid oxidation. In order to maintain quality and extend seafood shelf life, preservatives may be added during product processing and storage. As a response to consumer demand for fresh, minimally processed foods containing no chemical additives, the food industry has turned to natural preservatives in order to maintain the quality and safety of consumed foods. Natural preservatives should be effective against a broad spectrum of bacteria and fungi, be active at low concentrations, be nontoxic, should not affect food sensory properties, should impart no flavor or color to food and, finally, should be cost-effective. Natural preservatives may be isolated from microorganisms, animals and plants [25]. The main categories of natural preservatives used in seafood products include: organic acids, essential oils and plant/algal extracts, bacteriocins and chitosan.

#### 3.1.1. Organic Acids

Organic acids are compounds bearing one or more carboxyl groups (-COOH) in their molecule. In this form, they possess documented antimicrobial properties and exhibit hydrophobic characteristics, being soluble in the lipids of the cell membrane of microorganisms through which they enter into the cytoplasm. Within the cell, the high pH facilitates the dissociation of the acid. The accumulating hydrogen ions cannot exit through the cell membrane and, thus, cause acidification of the intracellular medium which, in turn, leads to the inhibition of enzymic reactions, creating a hurdle for microorganism growth and proliferation [32]. Factors affecting the antimicrobial activity of organic acids include: (i) the type of microorganisms (bacteria, yeasts, molds) against which the acid will have an activity, (ii) the polarity and the size of the molecule and (iii) the dissociation constant (pKa), indicating the form (dissociated or undissociated) of the acid enabling it to cross the cell membrane [33]. To ensure that organic acids dissociate at different pHs, a mixture of acids is often used. Such a mixture contributes to a synergistic effect. Another key feature of organic acids such as ascorbic and citric acids is that they function as chelating agents of metals, i.e., copper and iron, which exist in trace amounts in seafood flesh. Chelation of transition metals by organic acids renders the former unable to act as oxidation propagators. Organic acids have been approved as food additives, being categorized as generally recognized as safe (GRAS) [34].

##### Use of Organic Acids in Fish Preservation

Spraying and dipping are two methods that can be used to apply organic acids as food preservatives. Organic acids and their salts can inhibit bacterial proliferation of fish products. Garcia-Soto et al. [35] demonstrated the benefits of using lactic (0.5 g/L) and citric (1.25 g/L) acids in the icing agent of European hake and megrim to inhibit bacterial growth (aerobic, anaerobic, psychrotrophic, proteolytic and Enterobacteriaceae counts) as well as trimethylamine content. The authors concluded that an acid mixture icing medium is a good strategy in preserving the quality of fish. However, the direct addition of citric acid has been shown to negatively affect the sensory properties of products such as fish patties. To solve this problem, citric acid in the encapsulated form was used, thus maintaining sensory characteristics while sharply reducing lipid oxidation [36]. Sodium acetate, sodium citrate and sodium lactate have also been used for the inhibition of microorganism growth, the improvement of sensory properties and the shelf life extension of fish. Sallam [37] achieved an effective inhibition of spoilage microorganism growth by dipping salmon slices in aqueous solutions (2.5%) of sodium lactate, sodium acetate and sodium citrate. This led to a shelf life extension of salmon slices and delayed lipid oxidation during cold storage. Furthermore, organic acids and their salts have a pleasant taste and may be used for fish preservation via marination. Gokoglu et al. [38] used marinate solutions containing 10% sodium chloride and 2% or 4% acetic acid to dip sardine fillets for a period of 24 h. TMA-N and TVB-N values significantly increased during storage but remained within acceptable limits for 150 days. In contrast, sensory scores suggested rejection of samples after 120 days of storage. Rey et al. [39] studied the effect of a flake icing system composed of ascorbic, citric and lactic acids (400 and 800 mg/kg, C-400 and C-800, respectively) for the chilled preservation of hake, megrim and angler fishes. TVC recorded lower values for hake and megrim in the C-800 and C-400 batches compared to the controls. Regarding angler, lower values of TVC, psychrotrophs and proteolytic microorganisms were recorded for fish stored under the C-800 icing conditions. Sensory analysis revealed a considerable shelf life extension for fish treated with a mixture of ascorbic, citric and lactic (400 and 800 mg/Kg, respectively) acids. Kin et al. [40] monitored changes in microbial and quality parameters of catfish fillets in a brine solution with or without organic acid salts (potassium acetate and potassium lactate, 0.25–1.5%) packaged and stored at 4 °C for 14 days. After 7, 10 and 14 days of storage, untreated fillets exhibited higher psychrotrophic plate counts and increased spoilage compared to the fillets treated with potassium acetate and potassium lactate. Schrimer et al. [41] investigated the effect of a novel packaging method for fresh salmon. Fresh salmon was packaged in the presence of 20% CO_2_ and a brine solution containing various combinations of citric acid (3% *w*/*w*), acetic acid (1% *w*/*w*) and cinnamaldehyde (200 μg/mL). The use of (CO_2_ + organic acids) resulted in complete inhibition of bacterial growth for 14 days of storage at 4 °C. Addition of CO_2_, acetic acid and citric acid individually partly inhibited TVC, lactic acid bacteria, sulfur-reducing bacteria and Enterobacteriaceae. In all cases, inhibition effects were enhanced when combinations were used. Cinnamaldehyde had no effect on bacterial growth. Maqsood et al. [42] investigated the effect of catechin, caffeic acid, ferulic acid and tannic acid at various concentrations on the lipid oxidation of menhaden and mackerel mince. The most effective in retarding lipid oxidation was tannic acid, resulting in the lowest peroxide value (PV), conjugated diene (CD) and thiobarbituric acid-reactive substances (TBARS) values. This fact was partly related to the lower non-heme iron content in tannic acid-treated samples. On the other hand, the least effective in preventing lipid oxidation was ferulic acid. Melanosis, chemical, microbiological and physical changes in shrimp treated with catechin and stored in ice over a period of 10 days were monitored by Nirmal et al. [43]. Whole shrimp treated with catechin solution (0.05 or 0.1%) showed lower counts for psychrophilic bacteria, H_2_S-producing bacteria and Enterobacteriaceae during the entire storage period in comparison to controls. Lipid oxidation, loss of freshness and melanosis were reduced by catechin treatment. The effect of catechin was proportional to its concentration. The authors concluded that catechin can be used as a promising melanosis inhibitor as well as an antimicrobial and antioxidant in ice-stored shrimp. Lopez-Caballero et al. [44] used different melanosis-inhibiting formulations to treat frozen and re-thawed shrimp and monitored changes in several quality parameters during storage at 2 °C. Melanosis was retarded in the 4-hexylresorcinol-based formulations compared to untreated or sulfite-treated shrimp. The controls and sulfite treatment showed higher TVC, while lactic acid bacteria (LAB) was favored in the 4-hexylresorcinol and organic acid treatment. Finally, Monirul et al. [45] evaluated the efficacy of application of acetic acid and ascorbic acid spray for surface decontamination and shelf life extension of silver carp fish under refrigeration during a 9-day storage period. Fish fillets treated with the combination of acetic acid and ascorbic acid recorded a lower TVC, PV and pH than both untreated or individually treated samples with acetic acid or ascorbic acid. Sensory analysis showed that fish fillets with the combined treatment showed better quality retention after 9 days of storage.

#### 3.1.2. Essential Oils and Plant/Algal Extracts 

Essential oils (EOs) are plant oils containing a considerable number of odorous VOCs, produced as secondary plant-based metabolites, possessing several functional properties, including inhibition of germination and growth of pathogens. VOCs can be obtained by steam distillation from plants and usually possess antimicrobial and antioxidant properties [46]. EOs also have documented antimicrobial properties against various foodborne pathogens including *S. Typhimurium*, *E. coli O157: H7*, Campylobacter, *L. monocytogenes* and *S. aureus*. Factors influencing the efficacy of EOs include: their chemical structure, concentration, composition of the food matrix and method of application [47]. It has been widely accepted that, due to the hydrophobic nature of EOs, they interact with the bacterial lipid membrane, leading to an increased permeability of cell constituents which, in turn, results in cell death [48]. It has been shown that Gram-positive bacteria are generally more sensitive to EOs than Gram-negative bacteria [49]. EOs individually or in combination are often used in food preservation applications [25].

In turn, plant extracts (PE) are mixtures of phytochemical compounds extracted from plants using common solvents. These compounds are by-products of plant metabolism, produced as a defense mechanism and include: (i) terpenes, (ii) phenolic compounds and (iii) alkaloids [50]. Terpenes’ structure is made up of five carbons (C_5_H_8_)_n_ occurring as the simple or polymerized form. When they contain oxygen atoms, they are known as terpenoids. Phenolic compounds are made up of an aromatic ring containing one or more hydroxyl groups existing in either the monomeric form (phenols) or the polymeric form (polyphenols). Phenolic compounds include: flavonoids, hydroxybenzoic acids and hydroxycinnamic acids. Alkaloids are nitrogen-containing cyclic organic compounds synthesized from amino acids in the plant tissue. They include: caffeine, atropine and nicotine. [51]. Such bioactive phytochemicals occur in various parts of plants including leaves, stems, flowers, barks, seeds or roots. Being part of the plant defense mechanism, they are efficient in controlling foodborne pathogens and spoilage microorganisms. Lipophilic hydrocarbons such as terpenes and phenolics destabilize the cellular structure by solubilizing in the lipid bilayers of the plasma membrane and mitochondria. This leads to an increase in permeability of the cell membrane, resulting in loss of cellular constituents and disturbance of the active transport of substances [52]. The antioxidant properties of EOs and PE are mainly due to their high content of phenolic compounds. The hydroxyl groups of phenolic compounds provide hydrogen atoms to free radicals, inhibiting oxidation. EOs and PE are used as seafood preservatives at concentrations ranging from 0.1 to 1%. Higher concentrations, in most cases, negatively affect product sensory properties.

Macroalgae or seaweeds, classified as red (*Rhodophyta*), brown (*Phaeophyta*) or green (*Chlorophyta*), possess a number of beneficial constituents such as dietary fiber, amino acids, unsaturated fatty acids, vitamins and trace minerals as well as bioactive compounds such as polyphenols, carotenoids, alkaloids, phycocyannins and terpenes exhibiting antioxidant, antibacterial, antifungal, etc., activity [53]. According to the European Council regulation 258/97 [54], algae are considered as either food or food ingredients and thus can be used by the food industry without any hazard to health. In spite of the obvious advantages of algae as food preservatives, their use may be limited because of the flavor, odor and color they impart to foods, since effective doses to achieve preservation may exceed acceptable sensorial limits. A typical example of brown macroalgae, *Fucus spiralis*, has recently attracted significant attention because of its considerable nutritional content and the presence of various kinds of bioactive constituents with documented antioxidant and antimicrobial activity during fish preservation [55]. Furthermore, solvent selection becomes an important factor in the exploration of the specific activity of natural extracts because each type of solvent (water, methanol, hexane, etc.) has different extraction capabilities [56].

##### Use of Essential Oils and Plant/Algal Extracts in Fish Preservation

Goulas and Kontominas [57] investigated the effect of oregano essential oil (OEO, 0.4–0.8%) in combination with modified atmosphere packaging (MAP: 40% CO_2_/30% O_2_/30% N_2_) on the shelf life of lightly salted aquacultured sea bream fillets stored under refrigeration. TVB-N and TMA-N values of modified atmosphere (MA)-packaged fillets were significantly lower than their air-packaged counterparts. The degree of reduction in TVB-N and TMA-N values increased with increasing concentration of oregano oil. The salted samples were acceptable up to ca. 20–21 days, while the MAP plus OEO, salted samples were acceptable up to ca. 27–28 days of storage. Mexis et al. [58] investigated the effect of an O_2_ absorber in combination with oregano essential oil (0.4% *v*/*w*) on the shelf life of rainbow trout fillets stored under refrigeration. The O_2_ absorber and oregano oil inhibited microbial growth, which reached unacceptable levels (≥7 log cfu/g TVC) on the 4th and the 12th days of storage in the control and the O_2_ absorber and oregano oil groups, respectively. The fillets of the control group and the experimental group were organoleptically rejected on the 4th and the 17th days, respectively. TVB-N ranged between 10.6 and 54.6 mg/kg at the time of organoleptic rejection.

Attouchi and Sadok [59] added thyme powder (1% *w*/*w*) to fresh, ice-stored wild and farmed gilthead sea bream fillets in an effort to extend product shelf life. Lower values of TVB-N, TMA-N and TBA were recorded in thyme-treated fillets during ice storage. Thyme increased fillets’ shelf life by approximately five days. Gomez-Estaca et al. [60] wrapped cod fillets with gelatin-chitosan films containing clove and stored the products under refrigeration. Results showed a drastic reduction in Gram-negative bacteria, especially enterobacteria, while LAB remained practically unaffected for much of the storage period. Microbiological data were in good agreement with biochemical indices data, suggesting the potential of using these films for fish preservation. Ozogul et al. [61] added rosemary and sage tea extracts to vacuum-packaged sardine fillets stored at 3 °C for 20 days. The addition of rosemary and sage tea extracts resulted in lower ammonia and biogenic amine accumulation in sardine muscle. At the end of the storage period, putrescine and cadaverine contents of controls were 100-fold higher than those of treated groups. Attouchi and Sadok [62] added laurel and/or cumin EOs to fresh vacuum-packed (VP) wild and farmed sea bream fillets. The results showed that fillets with laurel or with cumin EOs recorded a lower TVC by ca. 0.5 to 1 log cfu/g and lower lipid oxidation by ca. 40%, extending the shelf life of fish fillets by approximately 5 days in ice storage. Li et al. [63] used tea polyphenols and rosemary extract to extend the shelf life of air-packaged whole crucian carp stored under refrigeration. Based on sensory analysis, the shelf life of crucian carp was found to be 7–8 days for controls, 13–14 days for the tea polyphenols-treated group and 15–16 days for the rosemary extract-treated group. Microbiological data were in good agreement with sensory data. Su et al. [64] investigated the antimicrobial effect of bayberry leaf extract (2 g/L) for the preservation of large yellow croaker. Results showed reduced bacterial growth in the treated group compared to that of the control group. Likewise, TVB-N, K-value and TBARS were significantly reduced compared to those of the control group. Houicher et al. [65] applied ethanolic extracts obtained from *Mentha spicata* and *Artemisia campestris* for the preservation of VP sardine fillets stored at 3 °C for a period of 21 days. The three groups tested were the control group, VM (group treated with 1% mint extract) and VA (group treated with 1% artemisia extract). The shelf life of sardine fillets was 10 days for control samples and 17 days for the combined treatment with mint and artemisia extracts. The treatment with natural extracts combined with VP retarded both microorganism growth and lipid oxidation, resulting in extension of product shelf life. Eskandari et al. [66] determined the antioxidant and the antibacterial activities of cumin and caraway extracts and their effect on the shelf life extension of silver carp, stored at 4 °C for 15 days. The results showed that both lipid oxidation and microbial spoilage of the samples were retarded in extract treatments compared to the control. Based on sensory analysis, the treatment with cumin extract resulted in a higher-quality product compared to the caraway extract. Results showed that both extracts provided a shelf life extension of fresh silver carp up to 6–9 days under refrigerated storage. Ga et al. [67] used red grape pomace extracts (RGP), rich in phenolic compounds with antioxidant and antimicrobial properties, to extend the shelf life of minced rainbow trout. Extracts were added to trout patties to give a final concentration of 0, 1 and 3%. RGP extracts delayed lipid oxidation and cadaverine formation in minced trout for 6 days under refrigeration. The authors concluded that RGP extract can enhance the quality and shelf life of minced trout patties while simultaneously providing a functional food with natural antioxidants beneficial to health. Kakaei et al. [68] used a biocomposite film of chitosan-gelatin (Ch-ge) containing 1% extract of grape seeds (GSE) and/or 2% *Ziziphora clinopodioides* essential oil (ZEO) extracted from blue mint bush and evaluated its potential to (i) control *Listeria monocytogenes* and (ii) extend the shelf life of minced trout fillets stored at 4 °C for 11 days. Essential oil analysis by GC-MS showed carvacrol (65.22%) and thymol (19.51%) to be the main components of the EO. Fish spoilage was significantly delayed in samples wrapped in the Ch-ge film containing different concentrations of GSE and/or ZEO in comparison to the control group. The lowest bacterial growth, PV and TVB-N content were obtained in fish samples wrapped in the film containing ZEO-2% + GSE-2%. The fillets treated with ZEO-2% + GSE-1% and ZEO-2% + GSE-2% received the highest sensory scores. Haute et al. [69] investigated the potential combination of cinnamon essential oil (CEO) and MAP to increase the shelf life of salmon. Salmon was dipped in a solution of 1% CEO, packaged and stored at 4 °C under vacuum or in 60% CO_2_/40% N_2_ modified atmosphere packaging (MAP). Results indicated that there was no benefit in the addition of CEO on microbial spoilage in salmon packaged under vacuum or in the tested MAP. Merlo et al. [70] used chitosan films containing pink pepper extracts and MAP (100% CO_2_) and monitored changes in the quality of filleted skinless salmon during refrigerated storage (2 °C) for 28 days. Two different treatments: chitosan film (CF) and chitosan film containing pink pepper residue extract (CFPP), were compared to the control. Results showed that CF and CFPP significantly reduced lipid oxidation relative to the control. Bacterial counts were significantly lower in CFPP, contributing to the significant reduction of trimethylamine. CFPP showed the lowest off-odor score. The results indicated that compared to CF, the CFPP extract film significantly improved the preservation and quality of refrigerated salmon fillets.

El-Sayed et al. [71] studied the effect of rosemary extract on the dynamics of microbial growth in smoked and non-smoked chilled Atlantic salmon packaged under MA. The authors observed significant antimicrobial activity of rosemary extract, reducing counts of *Bacillus cereus/thuringiensis* and *Citrobacter freundii* which dominated the microbiota of the controls. Hasani et al. [72] reported that pomace extract significantly delayed psychrotrophic bacterial growth as well as TVC and TVB-N of refrigerated silver carp. Maghami et al. [73] investigated the effect of chitosan nanoparticles (CNPs) loaded with fennel essential oil (FEO) in combination with MAP on microbial, chemical and sensorial properties of *Huso huso* fish fillets during refrigerated storage. Results showed that coating fish fillets with CNPs and FEO significantly reduced the PV, TVB-N and TBA values compared to the control samples. Microbiological analyses showed a lower value of TVC, psychrotrophic plate count (PPC), pseudomonas and LAB in coated fillets compared to control and MAP-treated samples. Fish fillets coated with CNPs and FEO showed high sensory acceptability throughout storage for 18 days. Hasani et al. [74] nano-encapsulated lemon essential oil (LEO) in chitosan/modified starch (Hicap) and investigated the antioxidant effect of the addition of 0.5 and 1% (*w*/*w*) free and nano-encapsulated LEO on the quality of fish burgers during storage up to 18 days. The addition of nanocapsules prepared by a mixture of CS/Hicap (1.5: 8.5% *w*/*v*) in LEOs significantly reduced PV, TBA and TVB-N values for all LEO nanocapsules-treated burgers in comparison to all other treatments. Sensory evaluation showed that the shelf life of burgers increased after incorporation of nano-encapsulated LEO. Ahmed et al. [75] investigated the preservative effect of garlic and ginger extract (GGE) on herring fish fillets stored at 4 °C for 8 weeks. GGE exhibited a considerable antioxidant and antimicrobial activity against *Bacillus subtilis*, *Clostridium botulinum*, *Escherichia coli*, *Salmonella senftenberg* and *Staphylococcus aureus*. The study reported that the GGE treatment resulted in significant microbial growth inhibition, reduction in lipid oxidation/TBA values and decreased protein oxidation. Furthermore, the GGE treatment preserved the sensory quality of the fish compared to the control for a period of 8 weeks. Barbosa et al. [76] investigated the stability against lipid oxidation of canned Atlantic chub mackerel. Two different concentrations of aqueous extracts of two abundant algae (*Fucus spiralis* and *Ulva lactuca*) were included in the brine packaging (aq. 2% NaCl) medium during mackerel canning. The product was stored at room temperature and sampled every 3 months. The loss of phospholipids, sterols and α-tocopherol, breakdown of fatty acids and PV of canned mackerel were partly inhibited in the presence of the most concentrated algal extracts. The study reported a preservative effect on lipid constituents and rancidity development in the presence of algal extracts in the packaging medium. Finally, Rebeiro et al. [77] evaluated the shelf life of frozen (−18 °C) minced tilapia by replacing synthetic preservatives (butylated hydroxy-toluene, BHT) with hijiki and nori red seaweed extracts. The chemical composition of the minced tilapia was not affected by application of the seaweed extracts, while TVB-N showed a lower increase rate in the presence of the extracts. The microbiological data complied with the Brazilian standards. Sensory evaluation showed no differences in the rancid aroma between treatments with natural and synthetic antioxidants used and only minor differences in the color of the products. The study concluded that the minced tilapia containing seaweed extracts was within national quality standards during frozen storage.

##### Use of Essential Oils and Plant/Algal Extracts in Fishery Products Preservation

Atrea et al. [78] evaluated oregano EO(OEO) and vacuum packaging (VP) for preserving Mediterranean octopus stored at 4 °C for a period of 23 days. Oregano-treated, VP octopus samples recorded significantly lower trimethylamine nitrogen and volatile basic nitrogen compared to controls. Furthermore, organoleptic evaluation indicated the beneficial effect of VP on the shelf life of octopus which increased with increasing concentration of OEO to 11 and 20 days for a concentration of 0.2 and 0.4% OEO, respectively. Xi et al. [79] evaluated the antimicrobial effect of phenolic compounds (at a concentration of about 4.6 g/L) extracted from tea on chilled-stored Pacific oysters. The authors observed a rapid and significant reduction in pathogenic *Vibrio parahaemolyticus* and the inhibition of TVC in Pacific oysters stored at 5 °C. The authors concluded that green tea extract may be used as a natural antimicrobial agent to inactivate pathogens, inhibit bacterial proliferation and increase the shelf life of refrigerated Pacific oysters. The antioxidant effect of mint and laurel essential oil extracts on organoleptic, microbial and biochemical changes of VP refrigerated eel (*Anguilla anguilla*) were investigated by Ozogul et al. [80]. The results indicated that both extracts inhibited oxidation of lipids and microbial growth in European eel, indicating their potential to extend the shelf life of this seafood product. Shiekh et al. [81] studied the inhibition of Pacific white shrimp polyphenoloxidase (PPO) using an extract of Chamuang leaves (CLE), characterized by high polyphenolic glycoside content, and organic acids (including 1,2-dihydroxy-1,2,3-propanetri- carboxylic acid and oxalosuccinate). CLE with copper chelation activity was effective in the inhibition of PPO. The magnitude of the effect was proportional to the concentration of CLE used. Compared to shrimp treated with 1.25% sodium bisulfite, shrimp treated with 1% CLE exhibited significantly lower melanosis over a period of 12 days under refrigeration. Likewise, CLE inhibited microbial growth, lipid oxidation and the elevation of TVB-N, compared to the other treatments. In conclusion, there was a significant effect on melanosis decrease and a range of other quality parameters of shrimp treated with Chamuang leaf extract.

#### 3.1.3. Biopreservation (Lactic Acid Bacteria, Bacteriocins)

Biopreservation usually refers to the application of naturally occurring microorganisms and/or their antimicrobial metabolites in order to preserve the quality of food products and to extend their shelf life [82]. Lactic acid bacteria have a major potential for use as biopreservatives as most are generally recognized as safe, and they naturally dominate the microflora of many foods. LAB compete for nutrients and produce numerous metabolites with antimicrobial activity such as organic acids (mainly lactic and acetic acids), antimicrobial peptides (bacteriocins), diacetyl and hydrogen peroxide which function as natural preservatives [83]. *Lactobacillus, Lactococcus, Pediococcus, Leuconostoc* and *Streptococcus* are the most important LAB genera for food preservation applications [84]. The acids are produced by the fermentative metabolism of LAB and may inhibit a large number of microorganisms due to pH reduction. The metabolites mentioned above are produced by LAB as a defense mechanism against the antagonistic microflora which compete for nutrients and oxygen. Among the above metabolites, hydrogen peroxide strongly oxidizes lipids of the cell membrane and destroys the molecular structure of cellular proteins [85]. Diacetyl functions by deactivating microbial enzymes by blocking or modifying catalytic sites [86].

Bacteriocins, among the metabolites produced by LAB, have documented biopreservative properties in foods due to their antimicrobial activity. At the same time, they do not affect the sensory attributes of foods [32]. Bacteriocins are actually small peptides or proteins with bactericidal or bacteriostatic action, synthesized in cell ribosomes [87]. They function by adsorption followed by penetration of the cell membrane through pores they create in the latter. Pores, in turn, cause an increase in membrane permeability, leading to the loss of cell constituents i.e., adenosine triphosphate ATP, amino acids, potassium and magnesium ions. Factors affecting the activity of bacteriocins include (i) interaction with proteins and lipids which can reduce bacteriocin activity, (ii) proteases which are able to inactivate bacteriocins, (iii) processing and storage conditions (pH, T), (iv) level of initial contamination of food (bacteriocins have a limited capacity in inhibiting large populations of contaminant microorganisms) and (v) composition of product natural microflora (bacteriocins inhibit only Gram-positive bacteria).

According to EU Directive 1129/2011/EC [88], the only bacteriocin presently approved for incorporation into food is nisin. As a biopreservative, nisin has been used to control the growth of Gram-positive bacteria such as *Staphylococcus aureus*, *Bacillus cereus, Listeria monocytogenes, Clostridium perfringens* and *Streptococcus* spp. [89]. Based on the factors limiting bacteriocins’ activity, nisin can be further stabilized through encapsulation in substrates such as chitosan or by using solid lipid nanoparticles or liposomes. In this method, a water-in-oil microemulsion is formed, in which EOs can be added to the oil phase, exhibiting a synergistic effect of two antimicrobial agents: nisin and EOs [90]. Furthermore, nisin-loaded microemulsions inoculated with rosemary, thyme and oregano EOs showed a bactericidal effect against *Bacillus cereus*, *L. monocytogenes*, *Staphylococcus aureus* and *Lactococcus lactis*.

Few studies have been carried out with the objective to extend the shelf life of seafood products using LAB and bacteriocins. The reason for this is the fermentative metabolism of LAB, resulting in the release of acids, carbonyl compounds, etc., which affect the sensory properties of fresh seafood products. Furthermore, while bacteriocins are effective against Gram-positive bacteria, the main spoilage microorganisms of most seafood are the pseudomonads and H_2_S-producing bacteria which are Gram-negative bacteria [91].

##### Use of Lactic Acid Bacteria and Bacteriocins in Fish and Fishery Product Preservation

Gomez-Sala et al. [92] tested the biopreservation potential of *Lactobacillus curvatus BCS35* on young hake and megrim. Several batches of fresh fish were inoculated with: (i) the LAB culture as a protective culture and (ii) its cell-free culture supernatant as a food ingredient. Fish were stored in ice within a chilled chamber at 0–2 °C at a retail fish market for 14 days. Microbiological analyses showed that treated fish had significantly lower bacterial counts compared to the untreated controls. The authors concluded that the biopreserved batches sold for a higher price in the fish market than the respective control batches. Anacarso et al. [93] investigated the potential of *Lactobacillus pentosus 39* to control the growth of *Aeromonas hydrophila* ATCC 14715 and *Listeria monocytogenes* ATCC 19117 inoculated into fresh salmon fillets at refrigeration temperatures. Results showed that the *Lb. pentosus 39* protective culture significantly reduced *A. hydrophila* and *L. monocytogenes* counts compared to controls. Ibrahim and Vesterlund [94] evaluated the inhibitory properties of 16 selected LAB and bifidobacteria against 32 spoilage organisms in VP raw Atlantic salmon. *Lactococcus lactis* subsp. *Lactis* proved to be the most effective inhibitory strain, resulting in a 3-day product shelf life extension compared to non-treated fish. At the same time, the addition of *L. lactis* did not alter the sensory and textural properties of the fish. The study showed that *Lactococcus lactis* may be used to increase the shelf life of VP raw fish stored at refrigeration temperatures. Sarika et al. [95] evaluated the biopreservative effect of *Lactococcus lactis* strain PSY2, isolated from the surface of marine perch, using fillets of reef cod. Fillets were sprayed with the bacteriocin solution, wrapped and stored at 4 °C. TVC was reduced by 2.5 log cfu/g units in the treated sample on day 14 of storage compared to the control. Sensory analysis showed a higher overall acceptability in the bacteriocin-treated samples stored for 21 days at 4 °C, while the untreated samples became unacceptable by day 14 of storage. Speranza et al. [96] inoculated anchovies kept in marinade brine (2% of acetic acid, 10% of NaCl and 200 ppm of citrus extract) with two probiotic strains (*Lactobacillus plantarum* and *Bidifobacterium animalis* subsp. *lactis*). Samples were packaged under different conditions (air, vacuum, in oil and in a diluted brine) and stored for up to three weeks at 4 °C. The limiting factors for shelf life determination proved to be sensory scores; the best sample was that packaged in diluted brine, retaining acceptable quality for three weeks. Selected LAB were used by Leroi et al. [97] to preserve cold smoked salmon (CSS) packaged under vacuum and stored at 8 °C. Specific spoilage organisms (SSO), *Photobacterium phosphoreum*, *Brochothrix thermosphacta* and *Serratia proteamaculans*, resulted in lower odor scores, whereas the spoilage potential of *Carnobacterium divergens* was weaker. *Lactococcus piscium* EU2241, *Leuconostoc gelidum* EU2247, *Lactobacillus sakei* EU2885 and *Staphylococcus equorum* S030674 were tested as biopreservative cultures. The protective effect of LAB differed from one SSO to another and no correlation could be established between sensory scores, SSO inhibition and acidification resulting from the protective cultures (PCs). Aymerich et al. [98] evaluated three potential bioprotective lactic acid bacterial strains against *L. monocytogenes* in three smoked salmon types with different compositional characteristics, primarily fat, moisture, phenol and acetic acid content. Of the three strains tested (*Lactobacillus sakei* CTC494, *L. sakei* CTC494 and *L. curvatus* CTC1742), *L. sakei* CTC494 inhibited the growth of *L. monocytogenes* after three weeks of storage at 8 °C in all the products tested. Results showed that this LAB strain may potentially be used as a bioprotective culture to improve the food safety of cold smoked salmon. Da Silva Vieira et al. [99] evaluated the effect of *Lactobacillus plantarum* on the preservation of fresh mussels. Mussels preserved with *L. plantarum* showed higher LAB counts and lower counts of *Vibrio* spp., as well as total heterotrophic bacteria, after 60 days of refrigerated storage. Saraoui et al. [100] selected two strains of lactic acid bacteria (*Lactococcus piscium* CNCM I-4031 and *Carnobacterium divergens* V41) for the preservation of peeled cooked shrimp (CPS). The latter proved very effective in retaining the sensory attributes of CPS. The panelists, however, perceived slight unpleasant odors and flavors due to the presence of *C. divergens* during the first 10 days of storage. In a mixed culture, *L. piscium* eliminated the off-odors and flavors released by *C. divergens*, while the co-culture maintained a good quality of CPS throughout storage. Therefore, a cocktail of the two cultures may be used as a strategy for the biopreservation of shrimp. Fall et al. [101] studied the antimicrobial effect of *Lactococcus piscium* CNCM I-4031 in cooked and peeled shrimp against *Brochothrix thermosphacta*. Shrimp were packaged under MA and stored at 8 °C. *Brochothrix thermosphacta* alone spoiled the product after 11 days, producing strong butter/caramel off-odors. In co-culture with *L. piscium*, sensory shelf life was extended by at least 10 days. The antimicrobial effect was partially explained by a drop in pH from 6.6 to 5.6. Matamoros et al. [89] isolated LAB from seafood products and evaluated their capacity to extend the shelf life of VP shrimp and cold smoked salmon. Different batches of cooked, peeled and VP shrimp were inoculated with seven LAB strains separately at an initial concentration of 5 log cfu/g, and degree of spoilage was evaluated by sensory analysis after 7 and 28 days of storage at 8 °C. The four strains showing the best results (two *Leuconostoc gelidum* and two *Lactococcus piscium* strains) were used for the same experiment involving cold smoked salmon. In this experiment, *Lactococcus piscium* strains showed higher inhibiting capacities, extending product sensory quality to 28 days of storage. Finally, Wiernasz et al. [102] investigated the use of six LAB strains, previously selected for salmon dill gravlax biopreservation. Salmon dill gravlax slices were inoculated by spraying with the protective cultures (PCs), reaching an initial concentration of 10^6^ log cfu/g. Samples were VP and stored for 25 days at 8 °C. PC antimicrobial activity was also assessed in situ against *L. monocytogenes*. Of the protective strains, *Carnobacterium maltaromaticum* SF1944, *Lactococcus piscium* EU2229 and *Leuconostoc gelidum* EU2249 dominated the microbial ecosystem and displayed antimicrobial activity against both the spoilage microbiota and *L. monocytogenes*. Of the three strains, *C. maltaromaticum* SF1944 was the most efficient in controlling *L. monocytogenes* growth. *V. fluvialis* CD264 was the only strain to extend the sensory quality, even beyond 25 days. This study concluded that *C. maltaromaticum* SF1944 and *V. fluvialis* CD264 both have a promising potential as bioprotective cultures to ensure salmon gravlax microbial safety and sensorial quality, respectively.

#### 3.1.4. Chitosan

Chitosan (CS) is a bioactive linear polysaccharide with documented antimicrobial properties. CS consists of polymeric 1→4-linked 2-amino-2-deoxy-β-d-glucose (acetylglucosamine and glucosamine sugar units). CS is commercially produced from chitin, obtained from exoskeletons of crustaceans and insects. The antimicrobial and structural characteristics of chitosan vary according to the levels of deacetylation which, in turn, can vary significantly according to the chitin source and the extraction procedure. Chitosan and its derivatives are biodegradable, biocompatible and nontoxic and have antioxidant and antimicrobial properties, rendering them valuable compounds for food and agricultural applications. Moreover, due to chitosan’s antimicrobial properties, CS-based coatings and films have been used for packaging food [103,104,105]. The antibacterial and antifungal properties of CS are mainly attributed to the interaction between the NH4^+^ groups of chitosan and the COO^−^ groups in the lipopolysaccharide layer of bacterial and fungal cell membranes. Such an interaction leads to changes in the permeability of the cell membrane, resulting in loss of its functional integrity, including leakage of bacterial enzymes and glucose. Factors which influence the cationic chitosan charge of CS include the degree of acetylation and pH. A high degree of deacetylation favors a higher concentration of positively charged amino groups. The pKa of amino groups (AGs) of CS is ca. 6.3. As a result, CS is protonated under slightly acidic conditions. On the contrary, at pH > 6.3, CS deprotonates and its antibacterial properties on microbial cell membranes are lost [106]. Other factors affecting the antimicrobial activity of chitosan include temperature, concentration and molecular weight. The viscosity and the molecular weight of CS may vary at elevated temperatures, affecting the stability and thus compromising antimicrobial activity. Extremes in molecular weight (1106 and 28 kDa) showed little or no antimicrobial properties of the polymer [107]. Chitosan also has antioxidant properties which are attributed to the binding capacity of amino groups to metal ions (Fe^2+^) which lead to stable macromolecular structures [108].

##### Use of Chitosan in Fish Preservation

Chitosan (CS) of different degrees of deacetylation and molecular weight is usually applied to seafood in the form of acid solutions at concentrations of 0.05–4%. Soares et al. [109] monitored microbial growth of CS-coated (1.5% solution) frozen salmon (*Salmo salar*, L.) for a period of 6 months. The antimicrobial effect of the CS coating was illustrated by the reduction in TVC, while TVB-N values remained stable during the experiment and the coating did not affect the texture of the samples. Analysis of organoleptic parameters suggested that chitosan was a better choice for frozen salmon samples, while in thawed and cooked samples, no significant differences existed between chitosan-coated and glazed samples. Tayel [110] investigated the preservative effect of fungal chitosan (M.W. 29 kDa, degree of deacetylation 91%) in processed fish sausages from Nile tilapia. Over a period of four weeks, refrigerated fish sausages supplemented with 1.5% chitosan exhibited sharp reductions in TVC, as well as total yeast and mold counts. A significant antibacterial effect of the CS coating was also reflected in the counts of other spoilage indicator organisms including Enterobacteriaceae and *S. aureus.* Likewise, sensory evaluation of the samples indicated better odor characteristics in stored chitosan-supplemented sausages compared to control samples. Saloco et al. [111] investigated the potential application of CS and maltodextrin (MD) encapsulated liquid smoke (LS) of coconut shells to inhibit microbial proliferation and lipid oxidation of fresh tuna. The authors reported a positive effect of this treatment in maintaining product quality when stored over a period of 48 h at ambient temperature. The concentration of nanocapsules did not affect product organoleptic scores. Mohan et al. [112] studied the effectiveness of an edible CS coating for the preservation of ice-stored Indian oil sardines. Compared to controls, which exhibited a shelf life of only 5 days, CS-coated samples maintained better textural properties and exhibited a lower level of lipid oxidation, bacterial load, TVB-N and TMA-N compared to controls. As a result, the CS-coated fish exhibited an increased shelf life (8 and 10 days for 1% and 2% CS-coated fish, respectively). Vatavali et al. [113] combined a CS coating with oregano EO for the preservation of red porgy stored in ice for 3 weeks. The product oxidative stability did not differ between control and treated groups, but the fish exhibited TVB-N levels above 30 mg N/100 g on days 13, 15 and 20 in the control, OEO-, CS- and CS-OEO-treated groups, respectively. Sensory evaluation indicated that the product was unacceptable on days 11, 16, 18 and 19 in the control, OEO-, CS- and CS-OEO-treated groups, respectively. Alak [114] studied the use of chitosan dissolved in acetic acid (AC) and lactic acid (LA) as a coating material for brown trout. The treatment with AC gave the lowest mean TVC, LAB, pseudomonads, pH, TBARS and TVB-N values. The authors concluded that the use of acetic acid in the chitosan film coating is more effective for the preservation of fish, compared to lactic acid. Duan et al. [104] vacuum-impregnated lingcod fillets with chitosan solutions containing krill oil. Fillets were then packaged under vacuum or MA and refrigerated (2 °C) for up to 3 weeks. The combination of chitosan treatment with either VP or MAP resulted in a reduction in TBARS, TVB-N and TVC values. The chitosan/krill oil coating did not affect the color of the fillets or consumer acceptance of both raw and cooked fish samples. Organoleptic evaluation indicated that chitosan-coated, cooked lingcod samples exhibited higher texture and odor scores compared to the control. Fan et al. [115] evaluated the effectiveness of a 2% CS coating in preventing spoilage of frozen silver carp (−3 °C). Microbiological, chemical and sensory evaluation results indicated the effectiveness of CS to prevent spoilage of frozen fish for over 4 weeks. Lopez-Caballero et al. [116] monitored changes in the quality of chilled cod patties coated with CS/gelatin or treated with fine CS powder. Spoilage was inhibited in the CS/gelatin samples. A reduction in the levels of total volatile basic nitrogen and the counts of Gram-negative bacteria was also observed. In contrast, no preservative effect was observed in using the CS powder in the patty mixture. Finally, Cao et al. [117] applied a chlorogenic acid (CGA) and chitosan (CS) coating to snakehead fish fillets. Treatments included: soaking of fish fillets in 2% chitosan (2CS), 0.2% CGA in 2% chitosan (0.2CGA/2CS), 0.5% CGA in 2% chitosan (0.5CGA/2CS) or 1.0% CGA in 2% chitosan (1.0CGA/2CS) solution. Coated samples were vacuum-packaged and stored at 2 °C for 5 months. Antimicrobial activity was non-significant among different treatments, while color, antioxidant and pH values were significantly different. Lipid and protein oxidation was inhibited in 2CS-, 0.5CGA/2CS- and 1.0CGA/2CS-coated fish fillets. Only the CS coating resulted in higher sensory scores and controlled browning. Considering antioxidant properties and other quality parameters, CGA/CS coatings may be applied commercially in fish preservation.

##### Use of Chitosan in Fishery Products Preservation

Chouljenko et al. [118] applied four different solution treatments to shrimp: The first solution was (AA) acetic acid (1%). The second solution was chitosan in acetic acid (CH). The third solution was sodium tripolyphosphate (TPP) in acetic acid solution. The fourth solution was a mixture of TPP + CH. Shrimp meat was separately vacuum tumbled with the solutions and frozen, and quality parameters were monitored over a period of 120 days. Controls included two groups: (a) shrimp meat that had been vacuum tumbled using distilled water and (b) shrimp meat that had not undergone vacuum tumbling. Shrimp treated using CH + TPP and CH had lower TVC as compared to other treatments used during the entire storage time. Treated shrimp were able to retain moisture, color and texture contents. CH + TPP and CH treatments produced the highest reduction in degree of lipid oxidation compared to other treatments. Moreover, the study also indicated that vacuum tumbling combined with CH + TPP or CH solution was effective in the reduction in TVC as well as lipid oxidation under frozen storage, while physicochemical properties were maintained. Carrion-Granda et al. [119] applied chitosan coatings, containing 0.5% of thyme and oregano EOs, onto peeled ready-to-eat shrimp tails packaged under MA and stored for 12 days at 4 °C. CH, serving as EO’s carrier, proved to be effective in inhibiting bacterial growth of the peeled shrimp. Sensory results indicated that both EO-containing coatings affected product sensory attributes. Chantarasataporn et al. [120] investigated the preservation of shrimp using chitin whiskers (CTWK), oligochitosan (OligoCS) and (CSWK) chitosan whiskers, all being nano-sized water-based chitin and chitosan derivatives. To maintain the color, quality and texture of the product stored at 4 °C, fresh shrimp were soaked in the additives for up to 48 h. The treatment at pH 8, which involved 0.25% of CSWK combined with 1% of NaHCO_3_ and 2.5% NaCl, proved to be the optimal condition that led to weight gain and a cooking yield that was as high as ~18% and ~14%, respectively. Yanar et al. [121] reported that commercially obtained CS or chitosan produced from *Metapenaeus stebbingi* shells was successfully used as a coating of refrigerated European eel. In fact, both chitosan additives significantly reduced the level of free fatty acid (FFA), PV and TBA. In an effort to extend the shelf life of Pacific oysters, Cao et al. [122] examined chitosan antimicrobial activity on oysters stored at 5 °C. The results revealed that *Vibrionaceae* (20%) and *Pseudomonas* (22%) were dominant microorganisms in raw oysters. The data also revealed that 5.0 g/L chitosan solution extended oysters’ shelf life from 8–9 to 14–15 days. In another study, Kucukgulmez et al. [123] used refrigerated storage to evaluate sensory, color and microbiological properties of European eel fillets using commercial chitosan and chitosan obtained from *Metapenaeus stebbingi* shells. The two CH solutions did not have any significant effect on the color parameters of the product during storage. However, controls showed higher a* values during the later stages of storage. The increase in TVC was slower in fillets containing chitosan compared to the control. The results showed that both chitosan additives maintained the quality characteristics of eel fillets and extended product shelf life during refrigerated storage.

### 3.2. High Hydrostatic Pressure 

High hydrostatic pressure (HHP) is a non-thermal method of preserving food, in which the product is processed under very high pressure, leading to the inactivation of microorganisms and enzymes in the food [26] (Figure 1).

The first food products preserved by pressure entered the Japanese market in 1990 [124]. HHP has a similar effect on microorganisms and enzymes to high-temperature treatment. Pressures applied during treatment are usually in the range of 100–600 Mpa but may be as high as 1200 MPa for spore inactivation (sterilization). Microorganism inactivation is the result of cellular damage and biochemical changes resulting from food exposure to high pressures. It has been shown that fungi are more susceptible to damage by HHP, followed by Gram-negative and Gram-positive bacteria [124]. Exposure to high pressure can also result in texture alteration of food products but such changes are reported to be reversible in the range of 100–300 MPa.

#### 3.2.1. Use of High Hydrostatic Pressure (HHP) in Fishery Products Preservation

Fish flesh is susceptible to thermal processing, affecting its textural and sensory properties. For this reason, HHP may be a suitable alternative to thermal processing of fish. Rastogi et al. [124] reviewed some important findings on the application of HHP to fish and shellfish products. Color changes and rancidity are side effects of exposing fish to high pressure. HP cold pasteurization or HP-assisted pasteurization results in lower levels of protein denaturation and enhanced organoleptic properties compared to conventional fish preservation methods. Sarika and Bindou [125] reported that compared to controls, tuna exposed to 220 MPa for a period of 30 min exhibited improved textural properties, inhibition of proteolysis and low levels of TVB-N and histamine. Similarly, Hogan et al. [126] reported that HHP at 200 MPa substantially increased the shelf life of yellowfin tuna chunks. Kaur and Rao [127] used HHP on tiger shrimp and observed that, compared to the controls, treatment at 435 MPa tripled product shelf life. Matejkova et al. [128] combined HHP with VP to preserve trout stored at 3.5 °C. The authors monitored the formation of biogenic amines which was inhibited in the HHP and VP-treated trout, resulting in the extension of product shelf life from 5–6 days to 3–4 weeks in the control and the HHP-VP fish, respectively. In contrast, the flesh of carp exposed to HHP exhibited a dull red color which can be attributed to the oxidation of myoglobin**** [129]. Yagiz et al. [130] used a range of pressures (150–600 MPa) to investigate the effect of HHP in preventing discoloration, microbial growth and rancidity in rainbow trout and mahi mahi. The authors observed optimal results with 300 MPa and 450 MPa in rainbow trout and mahi mahi, respectively. Erkan et al. [131] treated red mullet with HHP at 435 MPa for a period of 5 min at 3 °C. The HHP prolonged the shelf life of red mullet from 11 to 15 days at 4 °C storage. The shelf life of red mullet was further increased to 17 days, after 5 min exposure to 330 MPa at 25 °C. The effect of HHP on fish fillet quality was investigated by Erkan et al. [132]. Horse mackerel fillets were exposed to 220, 250 or 330 MPa for a period of 5 or 10 min at a range of holding temperatures. Compared to the controls, HHP did not affect the levels of TBA and TMA-N. Optimal results were obtained:(i)At 250 MPa, with 10 or 5 min holding at 7 and 15 °C, respectively;(ii)At 220 MPa, with 5 min holding at 15 or 25 °C;(iii)At 330 MPa, with 10 min holding at 25 °C.

Gunlu et al. [133] assessed the synergistic effect of HHP and VP in preventing spoilage of rainbow trout fillets. The authors reported a 4-day shelf life extension for vacuum-packaged HPP-treated fillets stored under refrigeration. Lee et al. [134] investigated the effects of HHP treatments on histamine-forming bacteria (HFB) *Morganella morganii* and *Photobacterium phosphoreum* in a tuna meat slurry using viability counting and scanning electron microscopy. The *D*-values of *M. morganii* (200 to 600 MPa) and *P. phosphoreum* (100 to 400 MPa) in the meat slurry ranged from 51.0 to 0.09 min and 71.6 to 0.19 min, respectively. *M. morganii* had higher *D*-values than *P. phosphoreum* at the same pressure value, indicative of its higher resistance to HHP treatment. The Zp values (pressure range that results in a 10-fold change in *D*-value) of *M. morganii* and *P. phosphoreum* were 153 and 105 MPa in the tuna meat slurry, respectively. Damage to the cell wall and cell membrane by HHP treatments was observed by scanning electron microscopy. Regarding shellfish, a reduction in target microorganisms by a factor of 10^5^ was achieved after application of high hydrostatic pressure (400 MPa) to oysters stored at 20 °C after 41 days [135]. Linton et al. [136] reported that when oysters underwent pressurization at 400 MPa, a reduced microbial count was recorded and samples did not spoil during a 4-week storage period. Similar results were obtained for mussels, prawns and scallops. Furthermore, Bindu et al. [137] showed that HHP helps in shucking the raw meat without cooking so as to remove the rigid shell of crustaceans and mollusks without losing the natural texture and appearance. In a study by Narwarkar et al. [138], clams were treated with HHP at various levels. The results indicated that a 90% reduction in TPC required a pressure of ≥480 MPa for HHP-treated clams. During sensory evaluation, panelists tasted both raw and processed clams, treated with HHP at 310 MPa pressure with 3 min holding time. There was no significant difference in the organoleptic scores of processed and raw clams. Koo et al. [139] treated raw oysters with HHP in the range of 230–586 Mpa and reported a reduction in *Vibrio* spp. up to 6 log cfu/g. Cruz-Romero et al. [140] studied the effect of a range of pressure values (260 to 600 MPa) on oysters, subsequently stored in ice for 4 weeks. All HHP treatments reduced microbial counts to undetectable levels. Small changes in color were observed during storage at 2 °C in ice, compared to untreated oysters. HHP increased lipid oxidation, at a rate proportional to the pressure applied. The study concluded that HHP processing can prevent spoilage of chilled-stored oysters, but it also affects product quality attributes. Ginson et al. [141] investigated the effect of HHP on decapitated Indian white prawns. The authors reported a significant effect of 250 MPa pressure (6 min holding time at 25 °C) on growth inhibition of bacteria, yeasts and molds. Compared to HHP-treated samples, controls exceeded the limit of acceptable TVC after twelve days of storage, whereas HHP-treated samples never reached this limit during the storage period investigated. The study showed that HHP processing can be used to prevent spoilage of Indian white prawns. Hughes et al. [142] used HHP at 100 or 300 MPa to treat post-rigor shucked red abalone. During cold storage at 2 °C for a period of five weeks, TVC did not exceed 10^6^ cfu/g and TVB-N levels remained below 35 mg/100 g for 35 days in the 300 MPa-processed samples. HHP did not affect sample color or textural properties. Both the control and 100 MPa-treated samples exhibited a significant content of biogenic amines. On the contrary, no biogenic amines were determined in samples processed at 300 MPa. The authors concluded that cold storage of abalone can be extended using HHP (by 4-fold compared to controls) without affecting product chemical or physical quality characteristics. Briones-Labarca et al. [143] treated red abalone with HHP (500 and 550 MPa for 8 min and 3 or 5 min, respectively) and evaluated its effect on product quality during cold storage at 4 °C. The results indicated that HHP reduced discoloration, whereas pH, water and ash content significantly increased; on the contrary, protein and fat were reduced. Over a period of cold storage for 60 days, total volatile basic nitrogen and trimethylamine levels remained well below 28 mg/100 g and 3 mg/100 g, respectively, in the HHP-processed samples. The control samples exceeded the limit of 30 mg/100 for TVB-N on day 30 of storage.

#### 3.2.2. Use of HHP in Controlling Pathogens in Seafood

Regarding the elimination of pathogens by HHP, Ritz et al. [144] reported that HHP at 200 MPa followed by freezing inactivated *L. monocytogenes* in smoked salmon. On the other hand, the treated product had a lighter color and increased toughness. Ye et al. [145] combined HHP followed by mild heating to inactivate *V. parahaemolyticus* and *V vulnificus* in vitro. Inoculated oysters were subjected to HHP at 200, 250 and 300 MPa, over 2 min at 21 °C. Subsequently, samples were exposed to mild temperatures (40–50 °C) for up to a period of 20 min. The authors observed a reduction in Vibrio counts in the HHP and heat-inactivated samples. The pathogens were absent in the samples exposed to pressure ≥275 MPa and subsequently to mild (45 °C) thermal treatment or at ≥200 MPa HHP, followed by thermal treatment at 50 °C for 15 min. Terio et al. [146] assessed the inactivation of hepatitis A virus (HAV) in Mediterranean (*M. galloprovincialis)* and blue (*M. edulis*) mussels. The authors monitored changes in HAV-contaminated mussels. HHP treatment at 300, 325, 350, 375 and 400 MPa for 5 min at ambient temperature significantly reduced the viral load by 1.7 log PFU/g (at 350) and by 2.9 log PFU/g (at 400 MPa) in *M. edulis*. Similar results were observed in *M. galloprovincialis.* Results demonstrated that HHP processing can successfully inactivate HAV in mussels. Phuvasate and Su [147] assessed the tolerance of different *Vibrio parahaemolyticus* strains to 200 and 250 MPa of HHP and subsequently investigated the effect of temperature and holding time on pressure-sensitive and pressure-tolerant strains of this pathogen. The authors concluded that the effectiveness of HHP in eliminating food pathogens requires an optimal combination of the level of pressure, thermal conditions and holding time. At 250 MPa, treatment effectiveness varied according to temperature used during HHP. At 5 °C, the reduction in *Vibrio parahaemolyticus* population ranged from 6.2 to above 7.4 log cfu/g for the tolerant of and the sensitive to pressure strains, respectively. The same treatment carried out at 1.5 °C reduced all strains to below the detection limit. Similar results were reported by Mootian et al. [148] regarding the effect of HHP on the inactivation of *V. parahaemolyticus* in live clams inoculated with up to 7 log cfu/g of a mixture of *V. parahaemolyticus* strains. HHP at 350 and 450 MPa for 6 and 4 min resulted in >10^5^ reduction in the population of *V. parahaemolyticus* to below the detection limit (<10 cfu/g). The study concluded that although HHP-processed clams may be free of *V. parahaemolyticus*, the effect on clam textural properties should be further studied. Ma and Su [149] investigated the potential of HHP for the inactivation of *V. parahaemolyticus* and shelf life extension of raw Pacific oysters using 293 MPa pressure, for 90–210 s at 8 °C. The authors reported a positive effect of HHP on shelf life extension from 6–8 days at 5 °C to 16–18 days in ice-stored samples. The authors concluded that HHP can be a part of the effective post-harvest practices used for the decontamination of *Vibrio parahaemolyticus* in raw oysters.

Human norovirus (HuNoV) infection of oysters poses a public risk. Leon et al. [150] investigated the potential of using HHP to inactivate HuNoV in oysters. The authors reported complete inactivation of HuNoV at 600 MPa at 6 or 25 °C. In summary, these data highlight the potential benefits of adopting HHP for shellfish, as a mitigation strategy for reducing HuNoV risk. Similar results were reported by Calci et al. [151] and Kingsley et al. [152], who demonstrated the inactivation of hepatitis A virus (HAV) in shellfish and oysters, respectively, using HHP. The effectiveness of HHP increased with increasing pressure.

### 3.3. Ozonation of Seafood

The sanitizing properties of ozone (O_3_) were firstly observed in 1909, in a meat storage plant in Germany, when it was observed that meat placed in proximity to an ozone generator, used to ventilate the storage container, exhibited reduced microbial load [153]. Since then, ozone has been used to reduce the microbial load and prevent spoilage of foods including seafood [154,155,156,157,158,159].

For industrial applications, including the food industry, O_3_ is produced by a corona discharge ozone generator [160]. When diatomic oxygen molecules (O_2_) are exposed to an electrical discharge of high-voltage alternating current, they are split into atomic oxygen (O) atoms which rapidly react with O_2_ to form O_3_ (Figure 2) [158]. Ozone is a highly unstable molecule, rapidly decomposing into reactive hydroxyl, hydrogen peroxide and superoxide ion radicals. The reactivity and great oxidizing power of ozone radicals make them a potent disinfectant both in the food industry and water treatment.

Ozone is a powerful disinfectant for bacteria, yeasts, molds, parasites and viruses. It causes extensive oxidation of (i) internal cellular proteins and (ii) unsaturated fatty acids in the cell envelope, resulting in rapid cellular damage. According to Khadre et al. [161], Gram-positive bacteria are more tolerant compared to Gram-negative bacteria.

Since 1997, the US Food and Drug Administration (FDA) has included O_3_ in the list of generally recognized as safe (GRAS) substances for use in food processing [162]. The degree of microbial inactivation achieved by treatment with ozone varies with pH, temperature, additives (i.e., sugars) and amount of organic matter surrounding the cells [161]. Furthermore, ozone either in aqueous and/or gaseous form has been recognized as a safe antimicrobial food additive by the USFDA [163]. It should be noted that when ozone is applied to food, it leaves no residues as it decomposes quickly [164]. However, due to its high reactivity, ozone usually promotes oxidation of food lipids, surface discoloration and often the development of off-odors in foods [161]. According to Goncalves [157], ozone application to fresh fish results in suppression of undesirable fishy odors, improving product sensory characteristics.

#### 3.3.1. Use of Ozone in Fish Preservation

It has been demonstrated that ozone treatment can effectively inhibit the spoilage of fish during cold storage. The effect of aqueous O_3_ on the shelf life of VP, cold-stored rainbow trout has been investigated by Nerantzaki et al. [156]. Ozonation treatment (for 60 or 90 min) reduced the bacterial load of fish as well as TVB-N and TMA-N compared to the control during storage. Furthermore, according to organoleptic evaluation and microbiological analysis, the ozone-treated samples exhibited a higher organoleptic score and extended shelf life. Gelman et al. [165] investigated the potential benefits of using ozone pretreatment of live tilapia before slaughtering. After slaughtering, the authors monitored bacterial, sensory, chemical and physical parameters of refrigerated fish. No significant effect of ozone treatment was observed when fish were stored at 5 °C; however, O_3_ treatment significantly extended the shelf life of tilapia by 12 days when fish were stored at 0 °C. Likewise, farmed turbot showed reductions in TVC when live fish or fillets were treated with ozone [166]. In a similar work, Campos et al. [167] investigated the potential benefits of using ozone in an ice slurry bath for the preservation of sardines. Organoleptic evaluation of the fish indicated that ozonated ice slurry prolonged product shelf life by 4 days and 11 days in comparison to sardines stored in ice slurry or ice flakes, respectively. The results of the organoleptic analysis were corroborated by monitored levels of TVB-N and TMA-N. Sopher et al. [168] evaluated the benefits of catfish ozone processing, starting from an aqueous pre-slaughter ozone treatment through the catfish filleting stage. The authors also investigated the effectiveness of gaseous ozone in removing odors in offal rooms of catfish processing waste. Results showed that aqueous ozone was very effective in reducing microbial loads on live catfish and their fillets. Gaseous ozone proved to be efficient in odor management in offal rooms. Aqueous ozone was also effectively used in trout processing by Dehkordi and Zokaie [169]. Aqueous O_3_, applied for two hours, inhibited microbial growth and the formation of PV and TVB-N. In turn, product shelf life was extended from 4 days in the control to 6 days in the O_3_-treated samples. There was no negative effect on organoleptic parameters of trout treated with aqueous ozone. The combined effect of the application of ozonated water pretreatment plus ozonated flake ice (OW + OIce) compared to traditional flake ice (CK), ozonated flake ice (OIce) and ozonated water pretreatment plus traditional flake ice (O) was investigated by Lu et al. [170] for the storage of Japanese sea bass. A slower increase in TVC, TVB-N and TBARS was observed for O + OIce compared to OIce, O and CK. The shelf life of Japanese sea bass treated with CK, OIce, O and O + OIce, determined by sensory evaluation, was 9, 15, 12 and >18 days, respectively. The results demonstrated superior effects of combining ozonated ice flakes and ozonated water on fish quality maintenance. Similar results were reported by Pastoriza et al. [171], who evaluated the combination of an initial immersion in ozonated water followed by immersion in ozonated ice flakes for preserving wild European hake subsequently refrigerated for a period of up to 18 days at 2 °C. Results showed that this method resulted in the extension of product shelf life by a week compared to the controls.

#### 3.3.2. Use of Ozone in Fishery Products Preservation

Manousaridis et al. [155] ozonated shucked mussels (1 mg/L) for 1 and 1.5 h, subsequently packaged under vacuum and stored under refrigeration. Ozonation decreased TVC, *Pseudomonas* spp., H_2_S-producing bacteria, *Brochothrix thermosphacta*, LAB and Enterobacteriaceae counts. The same decreasing trend was shown for TVB-N and TMA-N. Organoleptic analysis of the samples indicated that treatment with ozone for 1.5 h increased the shelf life of VP mussels from 9 in the control to 12 days in the ozonated mussels. Chawla et al. [172] reported that soaking shrimp in 3 ppm O_3_ for 1 h extended product shelf life based on bacterial loads. Soaking was more effective than spraying ozonated water in preventing spoilage. The antibacterial effect was proportionally increased with increasing ozone concentration and treatment time. Opkala [173] studied the quality attributes of ice-stored Pacific white shrimp subjected to minimal ozone treatment sequentially applied on the 1st, 3rd, 5th, 8th and 11th days of storage. The results indicated that TVC, TMA-N, TVB-N, *p*-anisidine value (AnV), total oxidation, titratable acidity (TA) and total color difference values changed significantly with storage. Significant reductions in total viable count, total volatile basic nitrogen, trimethylamine nitrogen and peroxide value in O_3_-treated samples compared to controls suggest ozonation as a promising technology for ensuring the safety of shrimp.

#### 3.3.3. Use of Ozone for the Decontamination of Seafood

*Vibrio parahaemolyticus* can cause foodborne disease outbreaks especially involving seafood products. Feng et al. [159] evaluated the following three parameters with regard to the efficacy of ozone treatment for the inactivation of *V. parahaemolyticus*: (i) aqueous O_3_ concentration; (ii) duration of treatment; and (iii) inoculated bacterial population. The most influential factor on the fate of the bacterium was shown to be the aqueous ozone concentration. Bacterial cell membranes remained intact at low O_3_ concentrations. On the contrary, at O_3_ concentrations above 1 mg/L, the functional integrity of bacterial membranes was compromised. Aqueous ozone penetrated the cells through leaking membranes, inactivating the enzymes and degrading the genetic material, eventually leading to cell death. The antibacterial efficacy of ozonated water (0.6–1.5 ppm) against *Listeria innocua* in seafood and on processing surfaces was evaluated by Crapo et al. [174]. Results showed that treatment with ozone drastically reduced the bacterial growth on stainless steel surfaces. The antibacterial efficacy of ozone was equal to chlorine. In contrast, O_3_ was not an effective bactericide in fish fillets and fish roe, while it increased lipid oxidation; in turn, ozone treatment resulted in a shorter product shelf life. The authors concluded that ozone may only be used as a sanitizer to clean seafood contact surfaces. Blogoslawski and Stewart [175] used ozonated saltwater as an antibacterial agent in aquaculture equipment and seafood products. After treatment with ozonated water, Vibrio counts were eliminated from the facilities of a shrimp farm. Ozonated ice was also effective in preventing spoilage of squid and salmon, extending product shelf life from 3 to 5 days. Ozone treatment was also successfully used to control the bacterial load of commercial ice-producing machines by a factor of 10^4^. Furthermore, no product flavor deterioration was noted using ozonated ice for preserving seafood. In a study by Louppis et al. [176], ozonation was used to reduce the levels of toxin in shucked mussels contaminated with diarrhetic shellfish toxins (okadaic acid, OA). Ozone treatment resulted in reduction in free OA which ranged between 6 and 100%, between 25 and 83% for OA esters and between 21 and 66% for total OA. Total content of diarrhetic shellfish toxins was substantially lower in homogenized mussel tissue compared to that of whole shucked mussels. Ozonated and control mussels exhibited a similar score of sensory evaluation and levels of TBA. The authors concluded that there is a potential benefit of using DSP detoxification of mussels, but further research is required for obtaining optimal results.

### 3.4. Irradiaton of Seafood

Ionizing irradiation is used as a food preservation method by the seafood industry to (i) extend product shelf life (by effectively destroying spoilage microorganisms), (ii) improve food safety (by destroying pathogens responsible for foodborne illnesses), (iii) delay or eliminate sprouting or ripening and (iv) control insects and invasive pests. Irradiation is achieved using gamma rays, electron beams or X-rays. The supplied energy abstracts electrons (ionizes) from atoms in the targeted food. Independent research carried out by the World Health Organization and food regulatory agencies in the USA and EU has confirmed that irradiation is safe [177,178,179]. Variations exist among different countries regarding regulations on which foods and at what doses can be irradiated. In numerous European countries, i.e., Austria, Germany and Greece, irradiation up to a dose of 10 kGy is only permitted for dried herbs, spices and seasonings. In contrast, in countries like Brazil and Pakistan, all foods are allowed to be irradiated. Irradiated food does not become radioactive (within the accepted energy limits, i.e., 10 MeV for electrons, 5 MeV for X-rays (US 7.5 MeV) and gamma rays from Cobalt-60), but irradiation can result in significant deterioration in the nutritional content and sensory properties of irradiated foods. Radiolytic products in the form of free radicals are another issue in food irradiation. The type of food and degree of treatment significantly affect changes in food quality and nutritional value caused by ionizing radiation [180]. Depending on the level of radiation that food is exposed to, there are three dose classes: (i) low- (≤1 kGy), (ii) medium- (1–10 kGy) and (iii) high-dose applications (>10 kGy) [181]. High-dose applications are currently not permitted in the USA for commercial food irradiation processing by the FDA and in the EU by EFSA. Irradiation treatments may be also be classified as radurization (≤1 kGy), radicidation (1–10 kGy) and radappertization (higher than 10 kGy).

Due to its penetration depth and uniform dose distribution, gamma irradiation can be used on a large scale and at a high volume. Treatment with electron beams (high-energy electrons) created within electron accelerators works for products that have low thickness as electron beams have a low penetration depth of a few centimeters [181]. Standards and regulations for the operation of irradiation facilities are covered by ISO 14470 and ISO 9001 [182]. According to the Codex Alimentarius, the FDA and EFSA, irradiated food or food products containing irradiated ingredients must be labeled as such, i.e., both the logo and statement “Treated with irradiation” or “Treated by irradiation” should appear on the food package.

Nowadays, food irradiation is widely applied to several types of food all over the world (spices, fruit, vegetables, meat and poultry). In the USA, for example, on a yearly basis, about 120,000 tons of food and feed destined for human and animal consumption, respectively, are irradiated [183].

About 30–35% of all seafood landed spoils before consumption due to inadequate processing and handling procedures. Thus, it would be logical to adopt any industrial process that would be able to destroy seafood spoilage microorganisms. However, this does not apply to irradiation which, although approved for use with a variety of foods by international organizations, still finds very limited applications in treating seafood. In Europe, a radiation facility has been established in Belgium which treats shrimp. In the US, the FDA has approved the irradiation of shrimp and prawns, as well as crab, lobster and crayfish, in order to eliminate or reduce foodborne pathogens and extend seafood shelf life [184].

#### 3.4.1. Use of Irradiation in Fish Preservation

In order to delay microbial spoilage and extend the shelf life of seafood under refrigeration, irradiation is usually carried out at doses of 1.0–3.0 kGy.

Chouliara et al. [185] monitored changes in VP, irradiated (at 1–3 kGy) sea bream samples stored under refrigeration. Sensory evaluation indicated that compared to the controls, a dosage of 3 kGy tripled the shelf life of sea bream. Mendes et al. [186] reported a 4 days longer shelf life of gamma-irradiated (at 1 or 3 kGy) ice-stored fresh Atlantic horse mackerel compared to controls. Silva et al. [187] assessed the effects of gamma radiation (1, 5 and 10 kGy) on ice-stored horse mackerel. The electrophoretic patterns of ice-stored horse mackerel muscle proteins was not affected by the γ-radiation applied, indicating the potential application of this method for fish preservation, provided that sensory evaluation of treated samples will show no adverse effects on product sensory attributes. Ozden et al. [188] determined the effect of γ-radiation (2.5–5 kGy) on the quality of refrigerated gilthead sea bream. The results indicated that irradiation extended the shelf life of this fish species with the effect increasing with the irradiation dose. Riebroy et al. [189] evaluated the effects of γ-radiation (up to 6 kGy) on the physicochemical properties, microbial quality and shelf life of a Thai fermented fish mince. The results showed that even though irradiation at 6 kGy inhibited microbial growth, it induced lipid and protein oxidation. Use of a dose of 2 kGy resulted in no negative effects on product quality for ca. 3 weeks. Mbarki et al. [190] studied the effect of γ-irradiation on lipid oxidation, microbial and physicochemical parameters of refrigerated iced bonito over a period of 3 weeks. The results indicated that spoilage microorganisms were eliminated at doses ≥1.5 kGy. The peroxide value increased with increasing radiation dose, indicating increased oxidation of lipids as a result of γ-irradiation. Based on microbiological, biochemical and textural properties, γ-irradiation at low doses extended product shelf life up to 3 weeks under chilled storage. Furthermore, the same authors [191] evaluated the effect of γ-radiation on ice-stored Mediterranean horse mackerel and reported a positive effect of the treatment on the quality index score using a 1 and 2 kGy dose. Compared to the controls, the treated samples exhibited a 5-day shelf life extension.

Altan and Turan [192] irradiated packaged bonito fish at 3 or 5 kGy. The product was subsequently frozen and kept frozen at −20 °C for over a year. Based on microbiological, chemical and sensory data, fish irradiated at 3 and 5 kGy and subsequently frozen retained acceptable quality for 12 months, while the control group was unacceptable after 9 months. Ozden et al. [193] irradiated sea bass at 2.5 and 5 kGy doses and evaluated the effect of irradiation on product quality and shelf life during ice storage. The authors reported a significant effect of irradiation on lowering the TVC and the levels of TVB-N. Likewise, compared to the controls, TMA-N and TBA values were lower in the treated fish. Sensory analysis indicated that irradiation prolonged the shelf life of ice-stored sea bass, by 2 and 4 days, at 2.5 kGy and 5 kGy radiation doses, respectively.

#### 3.4.2. Use of Irradiation in Fishery Products Preservation

Lee et al. [194] irradiated salted shrimp at 0, 5 and 10 kGy. One group was irradiated immediately after salting (15 and 20% salting), the other group was irradiated and subsequently fermented at 15 °C for 10 weeks. There were no significant effects of radiation on water activity, protein, lipid, moisture and salinity content. Shrimp irradiated and subsequently fermented exhibited a significant increase in TVB-N during fermentation, the increase being greater in the lower salt content group. However, this increase was reduced with increasing irradiation dose. On the basis of organoleptic parameters, microbial load and pH determined, the authors concluded that the combination of salting and irradiation was effective in the quality retention of fermented shrimp. Sharma et al. [195] irradiated whole or segmented shrimp at 2 kGy. Quantitative analysis of volatile components showed an insignificant effect of irradiation, suggesting that product sensory properties were also not expected to be affected. Sinanoglou et al. [196] assessed the effect of irradiation on the proximate composition and the fatty acid profile of frozen mollusks and shrimp. At a dose of 4.7 kGy, the total lipid content of both mollusks and shrimp decreased approximately by 6% compared to the controls. Irradiation also resulted in some qualitative changes in the fatty acid content, leading to a lower ratio of PUFA to SFA with increasing irradiation dose. Kim et al. [197] extracted the lipid fraction of dried squid and investigated the production of radiation-induced hydrocarbons and 2-alkylcyclobutanones. Such compounds were absent in non-irradiated squid and their concentration increased at doses >0.5 kGy. The authors concluded that radiolytic products of lipids, such as hydrocarbons or 2-alkylcyclo-butanones, may be used to monitor food safety for consumers, ensuring proper irradiation labeling of foods. A process was developed by Kannat et al. [198] for the preparation of shelf-stable, ready-to-eat shrimp, stored at room temperature, using a combination of hurdles including reduced water activity (0.85), packaging and gamma irradiation (2.5 kGy). Irradiation resulted in acceptable organoleptic and microbial quality of shrimp for over 8 weeks. On the contrary, controls developed mold growth within 15 days.

#### 3.4.3. Use of Irradiation for the Decontamination of Sea Food Products

Among microorganisms, bacteria in food can cause foodborne infection and intoxication in humans. Common seafood bacteria which can cause human infections include Vibrio, *Salmonella*, Shigella, *E. coli* and *L. monocytogenes.* Between 1973 and 2006, *Vibrio* spp. accounted for 54% of the illnesses related to seafood products. *Salmonella* and Shigella each were associated with about 10% of the reported illnesses, and *L. monocytogenes* with approximately 1%. Foodborne intoxications, on the other hand, occur when consumers ingest preformed toxins produced by rapidly growing bacteria present in food that has been inadequately processed. Botulism, for example, is a potentially fatal illness caused by a neurotoxin produced by *Clostridium botulinum* which grows under anaerobic conditions usually associated with vacuum-packaged, improperly canned or fermented products. Between 1973 and 2006, *C. botulinum* toxin was associated with approximately 25% of all reported seafood-related illnesses.

Several toxins produced by *Staphylococcus aureus* have also been implicated in foodborne intoxications. For example, *S. aureus* can produce heat-stable enterotoxins that cause foodborne illness, but less than 5% of the seafood-associated illnesses were associated with *S. aureus* during the above period of time. In order to prevent infection or intoxication during consumption of seafood, it is crucial to inhibit the growth of all above pathogens.

Besides bacteria, viral agents, i.e., norovirus, have been associated with seafood-borne diseases attributed to bivalve mollusks and finfish, causing 16% of all seafood-related outbreaks and almost 30% of the illnesses reported from 1973 to 2006. Hepatitis A, primarily associated with bivalve mollusks from polluted waters, is responsible for ca. 5% of all seafood-related outbreaks and illnesses [199].

As of 14 April 2014, the USFDA [200] has approved irradiation of crab, shrimp, lobster, crayfish and prawns to control foodborne pathogens and extend product shelf life. The approval refers to raw, frozen, cooked, partially cooked, shelled or dried crustaceans or cooked, or ready-to-cook, crustaceans processed with spices at the maximum dose of 6.0 kGy. Such an irradiation will reduce, but not entirely eliminate, the number of pathogens—including *L. monocytogenes, S. aureus,* Vibrio, *Salmonella*, Shigella and *E. coli*—in or on crustaceans. According to the FDA, the use of irradiation serves as a supplementary preservation technology not intended to replace stringent food safety standards that ensure the safety of seafood. Likewise, in Europe, EFSA has ruled on the food commodities allowed for irradiation including crustaceans and mollusks [178]. As of 2013, Crystal Seas Oysters LLC [201] began using a new irradiation facility in Mississippi, USA, to ensure that Vibrio is reduced to non-detectable levels in live oysters. *Vibrio vulnificus* and *Vibrio parahaemolyticus* are bacteria that occur naturally in warm coastal areas especially during the summer months, causing foodborne illnesses to consumers of raw or undercooked oysters, i.e., *V. parahaemolyticus* can cause non-bloody diarrhea as soon as 2 to 48 h after exposure. *V. vulnificus* infects the bloodstream of immune-compromised persons and after a 1 to 7-day incubation period, it can result in death within two days. Irradiation of seafood, such as warmwater shrimp/prawns and other shellfish, is carried out to improve their microbiological safety. Low doses (<3 kGy) eliminate 90–95% of spoilage microorganisms, resulting in an improvement in shelf life as well as all vegetative bacterial pathogens. Shrimp have a shelf life of 7 days when stored in ice. Treatment with 1.5 kGy extends product shelf life by ca. 10 days. A dose of 1 kGy eliminates both *E. coli* and *Vibrio* spp. in oysters without reduction in raw product quality. Oyster meat treated with 2 kGy has a shelf life of 21 to 28 days under refrigeration, compared to 15 days for their non-irradiated counterpart [202]. The Vibrios, most common in crustaceans and bivalve mollusks (*V. vulnificus* and *V. parahaemolyticus*), are very sensitive to irradiation, being reduced to below detectable levels with a treatment of only 300 Gy. *Salmonella* has frequently been found in farmed catfish samples obtained onsite and from retail markets in the USA [203]. The USFDA data from 1998 to 2004 regarding seafood imported to the US reported *Salmonella* contamination to be the most frequent contamination in catfish. Uncooked fish may contain *V. parahaemolyticus, Salmonella* spp. or *L. monocytogenes* [204]. Non-typhoidal *Salmonella* spp. in raw and RTE catfish are considered as high-priority microbial hazards [203].

According to Reed [205], a severe risk to human health may exist from the consumption of *Pangasius* spp. fish from Vietnam. This can be attributed to contamination of the fish with food bacterial strains of *Corynebacterium diphtheriae,*
*E. coli*, *Salmonella* spp., *V. cholerae* and *Cryptosporidia* spp. After thorough evaluation of *Salmonella* spp. present in seafood, FAO experts concluded that good hygienic practices during aquaculture production and biosecurity measures can minimize, but not eliminate, *Salmonella* in aquaculture products [206].

Shrimp is the most traded fishery product accounting for about 15% of the total international trade value of seafood products during 2012. It is therefore a widely consumed commodity which should be closely monitored to prevent foodborne diseases and prolong product shelf life [207]. Norhana et al. [208] reported that the prevalence and persistence of *Salmonella* and Listeria in shrimp and shrimp products (fresh and frozen) stresses the need for better control measures in order to eliminate these pathogens. Shrimp is frequently imported in the EU and the USA from several Asian countries which are major exporters of wild and farmed shrimp. There is some evidence to suggest that due to poor environmental conditions or hygienic issues at the processing sites, imports may occasionally not comply with mandatory microbial quality evaluation criteria set out for EU-producing countries or the USA. Pinu et al. [209] evaluated the microbiological quality of the frozen shrimp found in local markets and departmental chain shops of Dhaka city, Bangladesh. Most shrimp samples were heavily contaminated with pathogenic bacteria including Vibrio and *Salmonella*. Pathogenic bacterial load was found greater in samples obtained from large retail stores compared to local fish mongers. Asai et al. [210] analyzed samples of 29 types of seafood imported to Japan and reported that shrimp and prawns contribute to foodborne *Salmonella* infections.

Recently, safety risk studies associate the consumption of raw, undercooked or poorly processed fish and mollusks to foodborne illness outbreaks, especially when these commodities are often consumed raw. Olgunoglu [211] reported that the prevailing pathogens in seafood include *Vibrio* spp., *Salmonella* spp., hepatitis A, norovirus and parasites. Mollusks are organisms which feed on plankton suspended in the water they filter. The contamination of the aquatic environment of coastal ecosystems where mollusks are cultivated is a major issue, with frequently reported bacterial pathogens originating from agricultural or urban pollution.

Contamination of fishery products can also result at post-harvest stages during handling and transportation under inadequate conditions. Research in the EU on food-related outbreaks in 2014 showed that crustaceans, shellfish and mollusks were responsible for 8.1% of all outbreaks [212]. Based on the above data, several national and international food safety and public health agencies acknowledge the rising risk to the public from the consumption of raw fish and seafood products. For example, outbreaks of Salmonellosis are linked to eating raw or undercooked oysters.

Although the bacterial and viral load is effectively reduced by most fish processing methods (e.g., thermal treatment, ionizing radiation and HHP), eating raw oysters is hazardous. Bakr et al. [213] isolated *Salmonella* from 10% of samples (mollusks and crustaceans) from several retailers located in Alexandria, Egypt. Likewise, Brands et al. [214] found *Salmonella* in ca. 7% of oysters harvested in the USA, some of which contained *Salmonella enterica* serovar Newport, a strain which is resistant to multiple antibiotics and is responsible for outbreaks of human salmonellosis all over the world. Pathogens such as *V. parahaemolyticus, V. vulnificus,*
*L. monocytogenes* and several other bacteria which can be present in fresh or frozen mollusks and crustaceans can be inactivated by ionizing radiation, FDA permitting a maximum dose of 5.5 and 6.0 kGy, respectively [204]. In fact, most of the major pathogens will be inactivated at much lower doses of irradiation. For example, *Salmonella* isolated from grass prawns and shrimp homogenate is frequently reported to exhibit a D-10 value between 0.30 and 0.59 kGy. Consequently, irradiation can be effectively used to control major foodborne bacterial pathogens which pose a health risk to consumers.

Besides the inactivation of parasites and reduced bacterial load, irradiated fish and shellfish exhibit an extended shelf life. As for foods in general, irradiation is more effective in chilled compared to frozen products. In Europe, the EU Scientific Committee on Food (SCF) has recommended the doze of 3 kGy as sufficient to effectively reduce non-spore-forming bacterial pathogens by 2–5 log cfu/g for most of the fish and fishery products.

Jakabi et al. [215] assessed the effect of irradiation on organoleptic parameters and the viability of *Salmonella* strains in oysters inoculated with *S. enteritidis* and *S. infantis* serovars. Irradiation at 3.0 kGy did not affect oysters’ viability and organoleptic score but resulted in a significant reduction in both *Salmonella* strains by 5 to 6 logs. Sommers and Rajkowski [216] assessed the effect of deep freezing and irradiation on a *Salmonella* spp. cocktail isolated from shrimp and other seafood. The D-10 value of isolated *Salmonella* was 0.56 kGy. Irradiation with 2.25 kGy resulted in a significant reduction in *Salmonella* by 5 logs; the reduction was maintained in frozen samples for over a period of 3 months, indicating the effectiveness of this method in reducing the risk of salmonellosis from frozen shrimp. Louppis et al. [176] assessed the effect of gamma irradiation (6 kGy) and ozonation (15 mg/kg for 6 h) on the degradation of diarrhetic shellfish toxins of contaminated Mediterranean mussels harvested during two outbreaks of diarrhetic shellfish poisoning in Greece. Ozonated samples exhibited a reduced content of diarrhetic shellfish poisoning (DSP) toxins. Likewise, irradiated samples exhibited a reduced content of DSP toxins. Irradiation, however, negatively affected the appearance and texture of irradiated mussels.

Electron beam irradiation can be an effective method for reducing *L. monocytogenes* and spoilage bacteria on smoked salmon. Cold smoked fillets of Atlantic salmon were inoculated with *L. monocytogenes* and subsequently treated with e-beam radiation (at 1, 2 and 4 kGy). During 4 weeks’ chilled storage, irradiated samples exhibited significantly lower TVC and psychrotrophic bacterial load. The population of *L. monocytogenes* was reduced by 2.5 log cfu/g at 1.0 kGy and totally eliminated at doses equal to or above 2.0 kGy [217]. Robertson et al. [218] inoculated smoked mullet with a cocktail of *L. monocytogenes* strains (10^4^ cfu/g). Following VP, fish were X-ray-irradiated with doses which ranged from 0.5 to 2.0 kGy and stored at 3 °C and 10 °C for 90 and 17 days, respectively. X-ray irradiation had no effect on sensory flavor analysis and resulted in a significant reduction in *L. monocytogenes*, the effect increasing with increasing dose. A dose of 2.0 kGy completely eliminated the population of *L. monocytogenes* and the effect was maintained over the entire period of storage. Jo et al. [219] assessed the effectiveness of γ-irradiation on three ready-to-eat Korean seafood products inoculated with *S. Typhimurium, E. coli, S. aureus* and *L. ivanovii.* Products were inoculated and subsequently stored at 10, 20 and 30 °C for one day. The D-10 value of these pathogens ranged from 0.23 for *S. aureus* to 0.67 kGy for *L. ivanovii*. Irrespective of storage temperature, irradiation at 2.0 kGy eliminated the population of all pathogens apart from *L. ivanovii,* which required a higher dose (3.0 kGy) for its elimination. Collins et al. [220] applied electron beam irradiation to Eastern oysters inoculated with *Cryptosporidium parvum.* Treatment had no effect on the visual appearance of the irradiated oysters. Irradiated oyster tissues were fed to neonatal mice. Compared to controls, mice fed with oysters, irradiated at doses of 1.0 and 1.5 kGy, were much less susceptible to *C. parvum* infection. The infection was eliminated at 2 kGy.

### 3.5. Pulsed Electric Field Processing

Pulsed electric field (PEF) processing is a non-thermal food preservation technique used mainly for inactivation of microbes as well as in extraction, drying and other mass transfer processes. PEF technology consists of the application of short pulses of strong electrical currents with a short duration in the range of microseconds to milliseconds and intensity in the order of 10–80 kV/cm with the goal to inhibit microbial growth [221,222] (Figure 3).

When biological cells are exposed to pulsed electrical currents, the permeability of the cell membrane is affected, causing structural changes and local membrane breakdown. This phenomenon is reversible if the pores formed are small compared to the membrane area. Increasing the pulse width and/or number results in an increase in electric field strength (E) and treatment intensity, which, in turn, promotes the formation of large pores in the cell membrane. This causes irreversible damage to the cell membrane, leading to cell death [223]. The food product to be processed is placed in a treatment chamber where two electrodes are connected together with a nonconductive material to avoid electrical flow from one to the other. High-voltage electrical pulses are applied to the electrodes, which then conduct the high-intensity electrical pulse to the product placed between the two electrodes, causing, as mentioned above, microbial cell death. Compared to heat treatments, PEF offers several advantages as it can remove pathogens from unprocessed products without compromising their nutrient content and organoleptic properties. Pulsed electric field food processing is mostly used for the treatment of liquid and semi-solid food mixtures and for the extraction of food constituents [223]. The pulses of electric beams increase membrane permeability, enhancing, in turn, the efficiency of drying, extraction or diffusion processes involved in salting, marinating and other fish preservation methods. Exposure to PEF can inactivate parasites and also reduce the moisture content of a tissue. The latter is an important parameter for frozen products, as reduced moisture content reduces the formation of ice crystals and freeze damage. Pulsed electric field can also be used to enhance the extraction of food components with high nutritional value from fish processing by-products [224].

#### Use of Pulsed Light Technology to Fishery Products Preservation

PEF processing can be used to preserve the physicochemical properties and achieve desirable organoleptic parameters and nutrient and vitamin contents of the final product [225,226]. Klonowski et al. [221] presented some evidence to suggest that PEF treatment can render fish flesh more porous, increase water holding capacity and can be used as a pretreatment for fish drying. However, no improvement was observed on the tenderness of shellfish gastropod and mollusk products by these authors. Zhou et al. [227] assessed the effectiveness of PEF in extracting protein from mussels. The authors reported an extraction efficiency of 77.1% using 2 μs triangular PEF (20 kV/cm; 8 pulses and 120 min enzymolysis).

PEF processing has also been used for the valorization of fish by-products. A high-intensity pulsed electric field-assisted method for calcium extraction from fish bones was reported by Zhou et al. [228]. Compared to ultrasonic-assisted calcium extraction, pulsed electric field-assisted calcium extraction was more rapid and more efficient. In a similar study, PEF proved to be a rapid, efficient method of extracting chondroitin sulfate (ChS) from fish bones while reducing the waste product and potential pollution of chondroitin extraction [229]. He et al. [230] successfully combined PEF (22.79 kV/cm; 9 pulses) with semi-bionic extraction to improve the efficiency of extracting collagen calcium and ChS from fish bone. Li et al. [231] used pulsed electric field-assisted enzymic protein extraction in abalone visceral tissue. Optimal extraction was observed using 600 μs, 20 kV/cm and a 1:4 ratio of tissue to solvent. Compared to enzymic extraction, PEF-assisted enzymic extraction was more efficient and exhibited promising emulsifying properties. Nevertheless, the application of PEF resulted in lower viscosity and foaming properties of the extracted product. Furthermore, PEF processing failed in reducing enzyme activity of the fish. It should be noted that the electrical conductivity of the product is a crucial parameter that limits the application of PEF to materials with moderate conductivity [232,233]. In a study by Franco et al. [234], PEF processing was applied to extract antioxidants from three residues (gills, bones and heads) of two commercial species (sea bream and sea bass). Three methods of extraction using two solvents (water and methanol) and a water extraction assisted by PEF were assessed. Of the in vitro antioxidant methods used to evaluate the extracts, DPPH, ABTS and FRAP tests gave the highest antioxidant capacity values for residues from the sea bream species. In general, gills gave the highest antioxidant activity. Results suggest PEF as an environmentally friendly and economical method for the production of extracts with antioxidant activity from by-products of the fish industry.

### 3.6. Retort Pouch Processing (RPP)

Sterilization using heat is one of the most efficient methods of food preservation. The main objective of thermal sterilization is to kill all viable microorganisms including spores present in the food in order to achieve long-term shelf stability without the need for refrigeration. The application, however, of such severe heat treatments adversely affects the nutritional value of food including losses in vitamins and essential fatty acids and protein denaturation, particularly for products processed in metal or glass containers. Optimization of thermal processing conditions for minimizing nutrient loss without compromising the quality and safety of foods is a major issue for the food industry. One of the suitable options to overcome this problem is retort pouch processing [235,236,237].

The idea of retort pouches was promoted by the US army in the early 1950s and the research continued through the 1960s. It was finally invented by the United States Army Natick R & D Command, Reynolds Metals Company and Continental Flexible Packaging. These companies received the Food Technology Industrial Achievement Award for their invention in 1978.

Retortable flexible containers, usually in the form of pouches, are laminate structures that are thermally processed like a can or glass bottle. They are shelf-stable and can be stored at room temperature for a period of more than one year without the need for refrigeration. The most common form of pouch consists of a three-layer laminated material made (from outside to inside) of polyester/aluminum foil/cast polypropylene. Pouches made of polyester/aluminum foil/nylon/cast polypropylene are also available (Figure 4).
The polyester layer provides excellent strength and printability.The aluminum protects from exposure to light, gases, moisture and odors and prolongs product shelf life.The nylon layer protects from abrasion.The polypropylene layer acts as a heat seal surface and provides strength and flexibility.

Based on the above characteristics, it is noteworthy to mention that retort pouches possess good mechanical and heat transfer properties, high gas barrier and efficient sealing properties [238]. The materials that go into the packaging of retort pouches are FDA-approved and undergo sterilization processes which increase the durability of the packaging. Retort pouches and their types such as stand-up pouches, spout pouches and zip-lock pouches are also commercially available. The food to be thermally processed is first prepared, either raw or cooked, and then sealed into the retort pouch. The pouch is then heated to 116–121 °C for several minutes under high pressure inside a retort or autoclave. The food inside is cooked in a similar way to pressure cooking. The processing of foods in a retort pouch involves a series of operations including food product preparation, weighing, automatic transport to pouch, pouch opening, filling of product, pouch sealing, retort loading, retorting, retort unloading, drying and cartoning. The process is, in many respects, analogous to canning with the tin can being replaced by a cheaper heat-resistant flexible pouch. In comparison to frozen foods, the retort pouch provides a longer shelf life and does not require refrigeration, energy and expensive methods of distribution and storage. Major advantages of the retort pouch include [239]:The specific construction of the pouch provides rapid heat transfer for sterilization during processing. A 30–40% reduction in processing time is possible, with energy savings.Reduced heat exposure maintains product taste, color and flavor while resulting in fewer nutrient losses.Preparation of products that need to be heated to serving temperature can be accomplished in 3–5 min by immersing the pouch in boiling water or placing the plastic container in a microwave oven.Shelf life of retort pouch products is equivalent to that of foods in metal cans.Refrigeration or freezing is not required by packers, retailers or consumers.Pouches and containers do not corrode externally and there is a minimum of product–container interaction.Easy opening of the pouch.Empty retort pouches and nesting containers offer processors a reduction in storage space and lighter weight. Compared to empty cans, an equal number of retort pouches use 85% less space and are significantly lighter.Production of pouches uses less energy compared to metal containers.

#### 3.6.1. Use of Retort Pouch Processing in Fish Preservation

Simpson et al. [240] developed and optimized a mathematical model for thermal processing of conduction-heated foods in retortable pouches. The model was validated utilizing jack mackerel. The prediction errors obtained in the validation study were under 5%. Non-significant differences were found between the experimental and predicted values. Simulations showed that a significant reduction in process time (20–30%) could be attained utilizing variable retort temperature profiles while maintaining product quality. Manju et al. [241] conducted a study on seer fish moilee processed in retort pouches. Air inside the pouch was exhausted by steam injection, heat-sealed and heat-processed at 121.1 °C to a sterilizing effect (*Fo* 8.15 and total process time of 48.3 min. The shelf life of samples stored at ambient temperature (27 °C) was 18 months, whereas that of samples stored at 37 °C was only 10 months. Work carried out at the Central Institute of Fisheries Technology in India showed that sardines packaged in retort pouches made of polyester/aluminum foil/cast polypropylene had a shelf life of 3 years [242]. Kuda et al. [243] compared the quality of several retorted fish products treated with (i) the common retort (CR) process (using 115 °C, for 1.5 h) or (ii) with the high-temperature (125 °C) short-time (9 min) process (HTST). Analysis of the ATP-related compounds in raw fishes and retorted fish models showed that inosine monophosphate (IMP) was higher in HTST fishes than in CR fishes. In contrast, inosine (HxR), hypoxanthine (Hx) and K-value, an index of fish freshness, were higher in CR fishes. Sensory analysis showed that product umami and sweetness in the HTST fish were stronger than those of the CR fish. The bitterness was stronger in the CR fish compared to that of the HTST fish. The authors concluded that HTST is a favorable process for retorted fish products. The effect of thin metal oxide-coated barrier materials on the quality of shelf-stable salmon was investigated by Byun et al. [244]. Four different retort pouch structures were used: cast polypropylene (CPP); polyethylene terephthalate (PET)/silicon oxide-coated nylon/CPP (SiOx); aluminum oxide-coated PET/nylon/CPP (AlOx); PET/aluminum foil/CPP (FOIL). TBARS was measured during storage. Salmon packaged in SiO_X_ pouches had a higher TBARS value than salmon packaged in FOIL pouches after 8 weeks of storage. In sensory testing, salmon packaged in SiO_X_ pouches were less acceptable than salmon packaged in FOIL pouches after the same period. In contrast, salmon packaged in AlO_X_ and FOIL pouches had similar sensory and TBARS values. Overall, shelf-stable salmon packaged in AlO_X_ and FOIL had comparable shelf lives, while salmon packaged in SiO_X_ had a significantly lower shelf life compared to AlO_X_ or FOIL. Bindu et al. [245] prepared and processed fish peera, a traditional product from anchovies, in a retort pouch in an overpressure autoclave to an *F_o_* value of 7 and a cooking time of 66.02 min. Analysis of organoleptic, chemical and microbiological parameters showed that this method resulted in a shelf life of 1 year at a storage temperature of 28 °C. “Kalia”, a popular Rohu fish dish, was packaged in a four-layer laminated retort pouch and processed in a steam/air mixture overpressure retort at 121.1 °C to three different *F_o_* values of 7, 8 and 9 min [246]. Based on organoleptic and textural properties and the absence of viable microorganisms during storage, an *F_o_* value of 8 min with a total process time of 41.7 min at 121.1 °C was reported to be satisfactory for the preparation of Rohu fish curry in retort pouches. In a similar study, Majumdar et al. [247] packaged boneless fish balls from Rohu fish in retort pouches and processed them in an overpressure retort at 121.1 °C to three different *F_o_* values of 6, 7 and 9 min. Based on commercial sterility, sensory evaluation, color and texture profile analysis, an *F_o_* = 7 min and a total process time of 42.21 min at 121.1 °C were found to be satisfactory for maintaining product quality.

#### 3.6.2. Use of Retort Pouch Processing in Fishery Product Preservation

Bindu et al. [248] worked on the preservation of a ready-to-eat mussel meat product with the aim to retain its desirable sensory properties (natural texture and succulence). The product was vacuum-packaged and processed in a retort pouch in an overpressure retort. The total process time was 35 min to an *F_o =_* 9.8 and a cooking value of 90.3 min. Stored at room temperature, the samples exhibited a 1-year shelf life and achieved a high sensory score. Mohan et al. [249] prepared prawn “kuruma”, a dish based on Indian white shrimp. The product was packaged in conventional aluminum cans and in retort pouches. The retort pouch resulted in a 35.7% reduction in process time compared to aluminum cans of equal pack weight. Product sensory and textural attributes (color, firmness, chewiness and overall acceptability) in retort pouches were superior to those in aluminum cans. Mallick et al. [250] prepared shrimp in curry medium (SICM) and thermally processed the product in retort pouches to three different *F_o_* values, i.e., 5, 7 and 9. The respective cooking values obtained during the thermal processing of SICM were 59.20, 67.45 and 69.73 min. The sensory textural and color parameter values determined were in good correlation with those of the instrumental parameter values. The authors concluded that the study will help to standardize the *F_o_* value in order to achieve optimum sensory characteristics for the retort pouch-processed product.

Tribuzi et al. [251] processed chopped mussel meat packaged in retort pouches using a water immersion retort (*F_o =_* 7 min, retort temperature of 118 °C). Pretreated samples (salted and marinated) exhibited increased yield during 1-year storage at room temperature. There was no effect on the other physicochemical parameters. Freshwater prawn in curry was thermally processed by Majumdar et al. [252] to three *F_o_* values of 6, 8 and 9 at 116 °C. Respective process times were 53, 57 and 63 min, and the cooking value (CV) was 87.53, 107.93 and 117.55 min. Texture profile analysis showed that most textural parameters decreased with increasing *F_o_* values. On the other hand, color parameter values increased with increasing *F_o_* values. Optimum sensory scores were obtained when the product was processed to *F_o_* 7 min. Sreelakshmi et al. [253] developed a ready-to-eat sandwich spread from the meat of mud crab. The product was thermally processed in retortable pouches in an overpressure retort at temperatures of 111.1, 116.1 and 121.1 °C, to *F**_o_* values of 5, 6 and 7 min. The process was optimized by evaluating the samples for texture, color, commercial sterility, TBA value and sensory testing. All samples were found to be acceptable based on these tests. The sample processed at 116.1 °C for 6 min scored the highest, with a cooking value of 84.29 and a total process time of 42.59 min. A ready-to-eat thermally processed black clam product was developed by Bindu et al. [254], retaining its desirable natural texture and succulence. The product was vacuum-packaged in an in-house developed retortable pouch and processed in a still overpressure retort. The total process time was 44 min with an *F*_o_ value of 9 and a cooking value of 99 min. The product was rated as excellent by a sensory panel and had a shelf life of 12 months at ambient temperature (28 °C).

In conclusion, it should be noted that all above innovative processing technologies are applied in combination with innovative packaging technologies such as modified atmosphere packaging, vacuum packaging, active packaging, intelligent packaging and biodegradable packaging, both at the bulk and retail packaging levels. This final stage of processing will be dealt with, in detail, in a separate review article.

## Figures and Tables

**Figure 1 animals-11-00092-f001:**
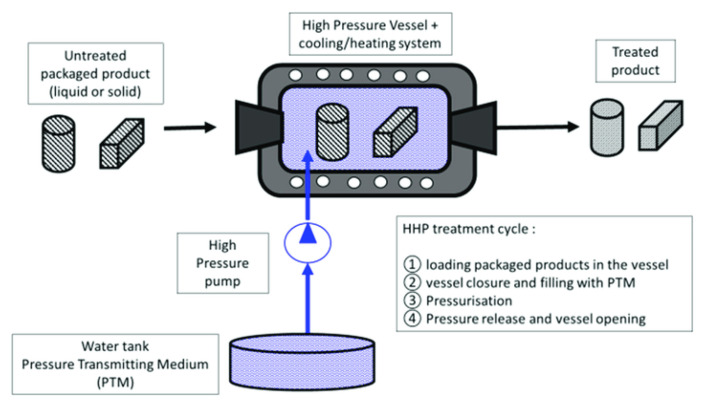
Schematic diagram of high hydrostatic pressure (HHP) processing.

**Figure 2 animals-11-00092-f002:**
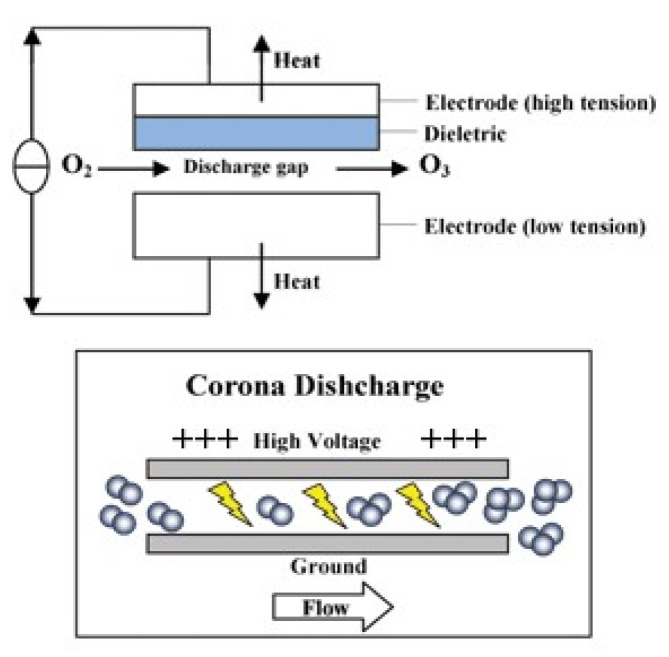
Scheme diagram of corona discharge method for the production of ozone.

**Figure 3 animals-11-00092-f003:**
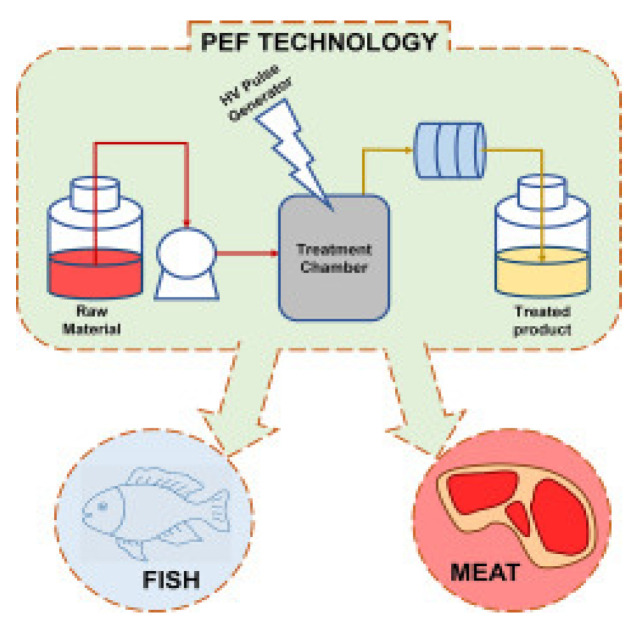
Principle of pulsed electric field (PEF) food processing (HV: high voltage).

**Figure 4 animals-11-00092-f004:**
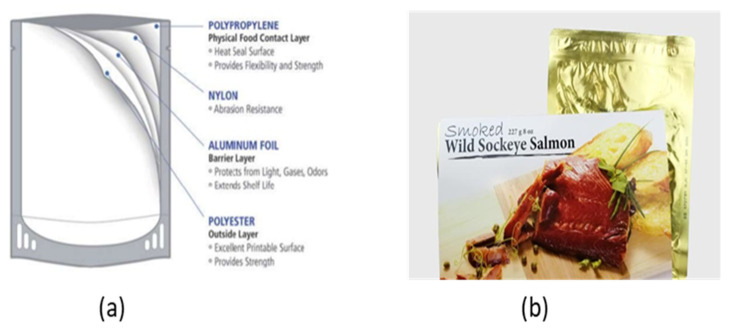
(**a**) Retort pouch structural construction (image courtesy: flairpackaging.com), (**b**) commercial retort pouch packaging for smoked salmon.

**Table 1 animals-11-00092-t001:** Proximate % composition of selected seafood products.

Seafood	Moisture	Carbohydrates	Proteins	Fat	Ash
**Bony fish**					
Bluefish	74.6	0	20.5	4.0	1.2
Cod	82.6	0	16.5	0.4	1.2
Haddock	80.7	0	18.2	0.1	1.4
Atlantic halibut	75.4	0	18.6	5.2	1.0
Atlantic herring	67.2	0	18.3	12.5	2.7
Atlantic mackerel	68.1	0	18.7	12.0	1.2
Pacific salmon	63.4	0	17.4	16.5	1.0
Swordfish	75.8	0	19.2	4.0	1.3
**Crustaceans**					
Crab	80.0	0.6	16.1	1.6	1.7
Lobster	79.2	0.5	16.2	1.9	2.2
Shrimp	72.5	0.9	20.5	5.5	0.8
Crayfish	80.0	0.5	17.0	1.5	0.9
**Mollusks**					
Clams, meat	80.3	3.4	12.8	1.4	2.1
Oysters	80.5	5.6	9.8	2.1	2.0
Scallops	80.3	3.4	14.8	0.1	1.4
Squid/mantle	83.5	1.4	13.5	0.8	0.7

Source: Watt and Merrill [4].

## Data Availability

Data sharing not applicable. No new data were created or analyzed in this study. Data sharing is not applicable to this article.

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
