# Peer review of "Innovative Seafood Preservation Technologies: Recent Developments"

_animals, 2021, doi:10.3390/ani11010092_

Round 1

Reviewer 1 Report

The review is very well organized and compiles most relevant information about the technologies currently available in the field of seafood processing.

Since the manuscript is so well structured and organized, I have no major concern about it. Just a couple of minor points:

  • please provide some information about the use of algae extracts in fish preservation.
  • English language and styling needs some polishing.

Author Response

Reviewer 1

-The review is very well organized and compiles most relevant information about the technologies currently available in the field of seafood processing.

Since the manuscript is so well structured and organized, I have no major concern about it. Just a couple of minor points:

Response: We thank the reviewer for his/her positive general comment

-please provide some information about the use of algae extracts in fish preservation.

Response: Information on the use of algal extracts in fish preservation has been included in the revised text. See revised text  l . 314, 346-359.

-English language and styling needs some polishing.

Response: We have made an effort to improve the English language throughout the text (see changes marked in red).

Reviewer 2 Report

The title of the work is "Innovative Seafood Preservation and Packaging
Technologies: Recent Developments".
The work is a review, covering 417 literature items, including 100 articles from 2015-2020.

General remarks:
1) Although the work is a review work, it should be shortened so that it focuses only on the topic of work. It did not cover issues related to food in general (including, e.g. meat).

2) The authors of the work should update the list of literature because according to the topic of the work, it should concern "Recent Developments".

Detailed comments:
Line 87-92. The authors present the composition without taking into account the proportion of fatty acids. It is known that polyunsaturated fatty acids (which seafood are rich in) are most prone to oxidation, especially when stored in aerobic atmospheres.

Extensive range of issues is not focused in detail on the results of the research (only exemplary fragments: lines: 1344-1350; 1388-1389; 2139-2198).
The results of the authors' research to be listed without any analysis.
Due to the extensive nature of the work and errors in it, the reviewer will only point out a small fragment of the work (from line 1362).

The results of research by various authors are presented in a general manner, without any summary, classification in terms of the influence of the type of atmosphere, storage time, temperature, barrier properties of the packaging to oxidation, e.g. white, lipids, sensory evaluation, microbiological changes.
There are no literature references, e.g. line: 1373, 1388.
Line: 1417-1418 - What was the storage temperature?
1430 - What was the composition of the modified atmosphere?
1437 - Should fish burgers be included in this part of the work or not in "Use of Modified Atmosphere Packaging for the preservation of fishery products"?
1445 - After what time the samples were not suitable for further storage?
2202 - What kind of atmosphere, what time of storage?

Section 4.2. (line 1688)  - there is a section on vacuum packaging.
So why is vacuum packing also included in section 4.1?

In the opinion of the reviewer, a thorough redrafting of the work is therefore necessary.

Author Response

Reviewer 2

General remarks:
1) Although the work is a review work, it should be shortened so that it focuses only on the topic of work. It did not cover issues related to food in general (including, e.g. meat).

Response: We agree that the text is extensive. However, it focuses only on seafood preservation and packaging and on no other food commodity

2) The authors of the work should update the list of literature because according to the topic of the work, it should concern "Recent Developments".

Response: The  reviewer is right. We have made an effort to update the literature, where possible, referring to most recent published work on the subject. See revised text l. 308-313, 440-473, 602-611, 681-689, 772-780, 1898-1903, 2198-2204, 2881-2883, 2898-2900, 2907-2909, 2954-2965, 3037-3040, 3121-3123, 3234-3235, 3343-3344, 3584-3585, 3605-3606, 3652-3654, 3677-3678, 3739-3743.  

Detailed comments:

Line 87-92. The authors present the composition without taking into account the proportion of fatty acids. It is known that polyunsaturated fatty acids (which seafood are rich in) are most prone to oxidation, especially when stored in aerobic atmospheres.

Response: It is clear that in this part of the text we refer to  fish muscle and not to fish lipids. The latter subject is treated in section 2.3. Oxidation and hydrolysis

Extensive range of issues is not focused in detail on the results of the research (only exemplary fragments: lines: 1344-1350; 1388-1389; 2139-2198).
The results of the authors' research to be listed without any analysis.

Response: we believe that further analysis to referred research work would have resulted to a considerably longer text. Furthermore, a more in depth analysis of findings would most probably not add substantially to the essence of a literature review.

Due to the extensive nature of the work and errors in it, the reviewer will only point out a small fragment of the work (from line 1362).

The results of research by various authors are presented in a general manner, without any summary, classification in terms of the influence of the type of atmosphere, storage time, temperature, barrier properties of the packaging to oxidation, e.g. white, lipids, sensory evaluation, microbiological changes.

Response: There are different ways of categorizing  information to be presented. We have chosen to classify information based on method of seafood preservation and packaging technology, which is perfectly legitimate. Had we included all the factors mentioned by the reviewer separately in the text the review would have been considerably more extensive, which contradicts the reviewer’s suggestion to shorten the text.

There are no literature references, e.g. line: 1373, 1388.

Response: We have added   the reference [257] now in L 1413 and reference [259] now in l. 1429 is [259] of revised text.

Line: 1417-1418 - What was the storage temperature?

Response: Atmosphere composition and temperature are now given in l. 1458.  

1430 - What was the composition of the modified atmosphere?

Response: L. 1430 (of initial text) now l. 1471 refers generally to the use of MAP and not a specific gas composition which differs from product to product.

1437 - Should fish burgers be included in this part of the work or not in "Use of Modified Atmosphere Packaging for the preservation of fishery products"?

Response: Under ‘fishery products’ we have included seafood other than fish i.e. crustaceans and mollusks. We have, thus, retained ‘fish burgers’ under the ‘fish preservation’ category

1445 - After what time the samples were not suitable for further storage?

Response: As we state in the text ‘Control  samples had a shelf life of 8 days, MAP samples 22 days and MAP (70 % CO2:30 % O2) plus sodium acetate (1%, w/v) 28 days’.

2202 - What kind of atmosphere, what time of storage?

Response:  In l. 2202 of initial text no reference was made to atmosphere and time of storage.  The reviewer is probably mixed up.

Section 4.2. (line 1688) - there is a section on vacuum packaging.
So why is vacuum packing also included in section 4.1?

Response: In case we refer to Vacuum packaging in section 4.1 (Modified atmosphere packaging) it is only because in specific work used  vacuum packaging was one of the treatments  assessed  along with Modified atmosphere packaging which was the major packaging treatment involved in the specific work.

In the opinion of the reviewer, a thorough redrafting of the work is therefore necessary.

Response: We believe that based on our system of information categorization, the work does not need redrafting.

Round 2

Reviewer 1 Report

The authors have properly addressed my criticisms. Accordingly, I recommend the publication of the revised manuscript in its present form.

Author Response

Reviewer 1 has already recommended the manuscript for publication.

Reviewer 2 Report

The reviewer maintains the position that the work should be shortened and redrafted.

The reviewer agrees with the authors that there are different ways to categorize the information presented. But the authors of the review should select the information so that it forms a coherent whole, and they did not do it (for example, the authors' answer regarding vacuum packing).

The reviewer notices a large amount of work involved in the preparation of this article, but its redrafting will undoubtedly contribute to increasing its scientific value.

Author Response

We are submitting the revised version of our manuscript: animals-923241 having deleted the subject of ‘Innovative packaging seafood technologies’. Thus, we have considerably shortened the text according to Reviewer’s 2 suggestion. Redrafting of the text was not attempted as this would basically require a completely new write up. We hope that this version will be satisfactory for your journal. We are also inclosing our last response to Reviewer No 2 in case the editor would like to be informed on the reviewing history of our manuscript.

Prof. Michael Kontominas

Reviewer No 2 latest comments

The reviewer maintains the position that the work should be shortened and redrafted.

The reviewer agrees with the authors that there are different ways to categorize the information presented. But the authors of the review should select the information so that it forms a coherent whole, and they did not do it (for example, the authors' answer regarding vacuum packing).

The reviewer notices a large amount of work involved in the preparation of this article, but its redrafting will undoubtedly contribute to increasing its scientific value.

Please find the cover letter attached.

Round 3

Reviewer 2 Report

I want to thank the authors for the changes introduced, which increased the work's value.
I suggest reviewing the work again because it contains numerous stylistic and punctuation errors (just a few examples below)

Line 14 -  unsaturated fatty acids (not fats).

Line 58 - it should be written "amino acids".

Line 59 - delete "and" 

Author Response

The changes suggested by the reviewer as well as other minor spelling errors have been corrected and shown in red.